# Ozone, DNA active UV radiation and cloud changes for the near global mean and at high latitudes due to enhanced greenhouse gas concentrations

Kostas Eleftheratos[1,2], John Kapsomenakis[3], Ilias Fountoulakis[4,5], Christos S. Zerefos[2,3,6], Patrick Jöckel[7], Martin Dameris[7], Alkiviadis F. Bais[8], Germar Bernhard[9], Dimitra Kouklaki[1], Kleareti Tourpali[8], Scott Stierle[10], J. Ben Liley[11], Colette Brogniez[12], Frédérique Auriol[12], Henri Diémoz[5], Stana Simic[13], Irina Petropavlovskikh[14], Kaisa Lakkala[15,16], Kostas Douvis[3]

[1]Department of Geology and Geoenvironment, National and Kapodistrian University of Athens, Athens, Greece
[2]Center for Environmental Effects on Health, Biomedical Research Foundation of the Academy of Athens, Athens, Greece
[3]Research Centre for Atmospheric Physics and Climatology, Academy of Athens, Athens, Greece
[4]Institute for Astronomy, Astrophysics, Space Applications and Remote Sensing, National Observatory of Athens (IAASARS/NOA), Athens, Greece
[5]Aosta Valley Regional Environmental Protection Agency (ARPA), Saint-Christophe, Italy
[6]Navarino Environmental Observatory (N.E.O), Messenia, Greece
[7]Deutsches Zentrum für Luft- und Raumfahrt, Institut für Physik der Atmosphäre, Oberpfaffenhofen, Germany
[8]Department of Physics, Aristotle University of Thessaloniki, Thessaloniki, Greece
[9]Biospherical Instruments Inc., San Diego, CA 92110, USA
[10]NOAA Global Monitoring Laboratory, Boulder, CO 80305, USA
[11]National Institute of Water & Atmospheric Research (NIWA), Lauder, New Zealand
[12]Univ. Lille, CNRS, UMR 8518 - Laboratoire d'Optique Atmosphérique, F-59000 Lille, France
[13]Institute for Meteorology and Climatology, University of Natural Resources and Life Sciences, Vienna 1180, Austria
[14]Cooperative Institute for Research in Environmental Sciences, University of Colorado, Boulder, CO, USA
[15]Space and Earth Observation Centre, Finnish Meteorological Institute, Sodankylä, Finland
[16]Climate Research Programme, Finnish Meteorological Institute, Helsinki, Finland

*Correspondence to*: Kostas Eleftheratos (kelef@geol.uoa.gr)

**Abstract.** This study analyses the variability and trends of ultraviolet-B (UV-B, wavelength 280–320 nm) radiation that can cause DNA damage, which are caused by climate change due to enhanced greenhouse gas (GHG) concentrations. The analysis is based on DNA active irradiance, total ozone, total cloud cover, and surface albedo calculations with the EMAC Chemistry-Climate Model (CCM) free running simulations following the RCP-6.0 climate scenario for the period 1960–2100. The model output is evaluated with DNA active irradiance ground-based measurements, satellite SBUV (v8.7) total ozone measurements and satellite MODIS/Terra cloud cover data. The results show that the model reproduces the observed variability and change of total ozone, DNA active irradiance, and cloud cover for the period 2000-2018 quite well according to the statistical comparisons. Between 50º N–50º S, the DNA-damaging UV radiation is expected to decrease until 2050 and to increase thereafter, as it was shown previously by Eleftheratos et

al. (2020). This change is associated with decreases in the model total cloud cover and negative trends in total ozone after about 2050 due to increasing GHGs. The new study confirms the previous work by adding more stations over low and mid-latitudes (13 instead of 5 stations). In addition, we include estimates from high latitude stations with long-term measurements of UV irradiance (3 stations in the northern high latitudes and 4 stations in the southern high latitudes greater than 55$^{\circ}$). In contrast to the predictions for 50$^{\circ}$ N–50$^{\circ}$ S, it is shown that DNA active irradiance will continue to decrease after the year 2050 over high latitudes because of upward ozone trends. At latitudes poleward of 55$^{\circ}$ N, we estimate that DNA active irradiance will decrease by 8.2 ± 3.8% from 2050 to 2100. Similarly, at latitudes poleward of 55$^{\circ}$ S, DNA active irradiance will decrease by 4.8 ± 2.9% after 2050. The results for the high latitudes refer to the summer period and not to the seasons when ozone depletion occurs, i.e., in late winter and spring. The contributions of ozone, cloud and albedo trends on the DNA active irradiance trends are estimated and discussed.

## 1 Introduction

The observed depletion of stratospheric ozone in the middle and high latitudes in the 1980s and the 1990s was followed by a general stabilization in the 2000s and by signs of recovery in the 2010s (Solomon et al., 2016; Weber et al., 2018; Krzyścin and Baranowski, 2019). The general behavior of ozone in the last 4 decades motivated research into the response of UV variability to ozone variability during periods with and without ozone decline. UV-B radiation is of special importance because of its effects on human health and the environment. In the short-term, the biological effects of UV-B radiation on humans include skin effects (erythema, photodermatitis) and eye effects (keratitis, conjunctivitis). Long-term effects include skin cancer, skin aging and cataracts. UV radiation can also damage the immune system and DNA (Lucas et al., 2019, Section 3.2 and references therein).

Changes in UV-B radiation and their relation to the depletion of the ozone layer in the stratosphere are being studied since the early 1990s (e.g., Blumthaler and Ambach, 1990; McKenzie, 1991; Bais and Zerefos, 1993; Bais et al., 1993). Early measurements of solar UV irradiance suggested that the long-term increase of the strongly ozone dependent wavelength of 305 nm was solely attributed to the observed stratospheric ozone decline and that it was not the result of improvements of air quality in the troposphere and changes in environmental conditions (Kerr and McElroy, 1993; Zerefos et al., 1998). Later studies based on longer atmospheric measurements looked at the effects of cloud cover, aerosols, air pollutants and surface reflectance on the long-term UV variability (e.g., Bernhard et al., 2007; den Outer et al., 2010; Kylling et al., 2010; Douglass et al., 2011). Over Canada, Europe, and Japan, it was found that the observed positive change in UV-B irradiance could not be explained solely by the observed ozone change and that a large part of the observed UV increase was attributed to tropospheric aerosol decline, the so-called brightening effect (Wild et al., 2005), since cloudiness had no significant trends (Zerefos et al., 2012). At

high latitudes on the other hand, it was found that the long-term variability of UV-B irradiance was not
affected by aerosol trends but by ozone trends (Eleftheratos et al., 2015).
Further efforts to understand the interactions between solar UV radiation and related geophysical variables
were done by Fountoulakis et al. (2018). They concluded that the long-term changes in UV-B radiation
vary greatly over different locations over the Northern Hemisphere, and that the main drivers of these
changes are changes in aerosols and total ozone. Updated analysis of total ozone and spectral UV data
recorded at four European stations during 1996–2017 revealed that long-term changes in UV are mainly
driven by changes in aerosols, cloudiness, and surface albedo, while changes in total ozone play a less
significant role (Fountoulakis et al., 2020b). Over higher latitudes, part of the observed changes may be
attributed to changes in surface reflectivity and clouds (Fountoulakis et al., 2018 and references therein).
Dedicated studies assessing trends of UV radiation in Antarctica provided further evidence that the UV
indices are now decreasing in Antarctica during summer months, but not yet during spring when the ozone
hole leads to large UV index variability (Bernhard and Stierle, 2020). The downward trends in UV index
during summer are associated with upward trends in total ozone.
Long-term predictions of UV radiation are governed by assumptions about the future state of the ozone
layer, changes in clouds, changes in tropospheric pollution, mainly aerosols, and changes in surface albedo.
Unpredictable volcanic eruptions, increasing emissions of GHGs, effects from growing air traffic, changes
in air quality and changes in the oxidizing capacity of the atmosphere induce uncertainties to long-term
predictions of ozone and therefore to UV radiation levels (Madronich et al., 1998). The Environmental
Effects Assessment Panel of the United Nations Environment Programme publishes the most recent global
environmental effects from the interactions between stratospheric ozone, UV radiation and climate change.
The Panel noted that future changes in UV radiation will be influenced by changes in seasonality and
extreme events due to climate change (Neale et al., 2021). Simulations of surface UV erythemal irradiance
by Bais et al. (2011) showed that UV irradiance will likely return to its 1980 levels by the first quarter of
the 21st century at northern mid and high latitudes, and 20-30 years later at southern mid- and high
latitudes. After reaching this level, UV will continue to decrease towards 2100 in the Northern Hemisphere
because of the continuing increases in total ozone due to circulation changes induced by the increasing
GHG concentrations, whereas it is highly uncertain whether UV will reach its 1960s levels by 2100 in the
Southern Hemisphere (Bais et al., 2011). However, in the Arctic, large, seasonal loss of column ozone
could persist for much longer that commonly appreciated. Projections of stratospheric halogen loading and
humidity with General Circulation Model (GCM)-based forecasts of temperature, suggested that conditions
favorable for large Arctic ozone loss could persist or even worsen until the end of this century, if future
GHG concentrations continue to steeply rise. Consequently, anthropogenic climate change has the potential
to partially counteract the positive effects of the Montreal Protocol in protecting the Arctic ozone layer
(von der Gathen et al., 2021). CCM simulations of DNA-damaging UV variability analyzed by Eleftheratos
et al. (2020) showed that UV irradiance will likely increase at low and mid-latitudes during the second half
of the 21st century due to decreases in cloud cover driven by climate change caused by enhanced GHG
concentrations.
GHG changes can be an important driver of cloudiness changes. Norris et al. (2016) provided evidence for
climate change in the satellite cloud record. They estimated fewer clouds over the mid-latitudes from 1983
to 2009 and concluded that the observed and simulated cloud change patterns are consistent with poleward
retreat of mid-latitude storm tracks, expansion of subtropical dry zones, and increasing height of the highest
cloud tops at all latitudes. The primary drivers for these changes were found to be the increasing GHG
concentrations and a recovery from volcanic radiative cooling (Norris et al., 2016). In the same direction,
Schneider et al. (2019) showed that stratocumulus clouds, some of the planet's most effective cooling
systems, become unstable and break up into scattered clouds under increasing GHG concentrations. Their
results also showed that less clouds will trigger additional surface warming in addition to that from the
rising CO2 levels (Schneider et al., 2019). Both studies provided indications that increasing GHGs can
affect clouds, which in turn can affect the UV radiation changes.
In this work we investigate the UV variability and trends for the near global mean (50° N–50° S) and at
high latitudes due to the expected increase of GHG concentrations in the future. We show that DNA active
irradiance will continue to decrease after 2050 at high latitudes due to the prescribed evolution of
greenhouse gases in contrast to regions located between 50º N and 50º S where it is shown to increase. The
year 2050 was chosen as a mid-point to evaluate the trends as it divides the 21st century into two equal
periods, 2000-2049 and 2050-2099, but most importantly because it was noted that for a Representative
Concentration Pathway (RCP) of 6.0, the Chemistry-Climate Model Initiative (CCMI, phase 1) simulations
project that global total column ozone will return to 1980 values around the middle of this century (Dhomse
et al., 2018). Our study confirms the previous work by Eleftheratos et al. (2020), which focused on ozone
profiles from five well-maintained lidar stations at low and mid-latitudes. Here, we add more ozone and
UV stations in mid-latitudes and include estimates from high latitude stations with long-term measurements
of UV radiation. The analysis aims to investigate whether the increase of DNA active radiation predicted
for mid-latitudes in view of climate change, will also be observed at high latitudes. To address the issue, we
use the same methodology as Eleftheratos et al. (2020), in which we compare two CCM simulations; one
with increasing GHGs according to RCP-6.0 and one with fixed GHGs emissions at 1960 levels. The
variability of ozone from the model simulations is evaluated against solar backscatter ultraviolet radiometer
2 (SBUV/2) satellite ozone data. The variability of DNA active irradiance from the model simulations is
evaluated against ground-based DNA active radiation measurements, and the variability of simulated cloud
cover is evaluated against cloud fraction data from the MODerate-resolution Imaging Spectroradiometer
(MODIS)/Terra v6.1 satellite dataset.
It is important to clarify the novelty of this research and its added value with respect to the previous study,
and to point out the main differences and similarities. The objective of this research is to study how total
ozone, DNA active irradiance and cloud cover might change in the future at high latitudes due to the
increasing GHGs in comparison to the near global mean (50º N–50º S). Also, to estimate the part of the
DNA active irradiance change that can be explained by ozone and cloud changes in the future using
multiple linear regression (MLR) statistical analysis. The previous study by Eleftheratos et al. (2020) did
not look at high latitudes and did not apply MLR analysis to quantify the related contributions to the DNA
weighted UV irradiance. The previous work analysed 5 stations between 50º N and 50º S. The new study is
enriched with more stations at the near global scale (13 instead of 5 stations, of which 4 are identical) and,
in addition, it includes analysis of averages in latitudinal bands, which was not done in the previous study,
thus providing more complete results.
The study is organized as follows. Section 2 describes the data sources and methodology. Section 3 shows
the variability and projections of DNA-damaging UV radiation at high latitude stations in comparison to
mid-latitude stations, and, finally, Section 4 summarizes the main results.
**2 Data sources**
**2.1. Ground-based data**
We have analyzed DNA-weighted UV irradiance data at 20 ground-based (GB) stations listed in Table 1.
Although the DNA action spectrum tends to exaggerate UV effects on humans, mammals, etc., (as it was
determined with bacteria and viruses and does not take the wavelength dependence of the skin's
transmission into account), it is the appropriate action spectrum for studying the detrimental biological
effects of solar radiation and the effective dose of UV radiation in producing skin cancer (Setlow, 1974).
Most of the stations listed in Table 1 contribute spectral UV data to the data repository of the Network for
the Detection of Atmospheric Composition Change (NDACC, www.ndacc.org) at https://www-
air.larc.nasa.gov/pub/NDACC/PUBLIC/stations/ (last access 4 September 2022) (De Mazière et al., 2018)
and have been reported among those possessing high-quality long term UV irradiance measurements
(McKenzie et al., 2019). Sites not part of NDACC are Aosta, Athens, Sodankylä, and Thessaloniki. Data
from these stations are of high-quality as well (e.g., Fountoulakis et al., 2018; Fountoulakis et al., 2020a;
Kosmopoulos et al., 2021; Lakkala et al., 2008). The high quality of the spectral UV measurements is
ensured by applying strict calibration and maintenance protocols.
We have calculated monthly mean irradiances from noon averages for all stations listed in Table 1 (average
of measurements ± 45 minutes around local noon) and compared them with the DNA active irradiance data
from an EMAC sensitivity simulation (internally named SC1SD-base-02), with specified dynamics
representing the recent past (2000-2018) as a means for model evaluation. The comparisons are presented
in Section 3.1 and in the Supplementary materials of this study for each station separately.

## 2.2. Satellite data

We have analyzed the daily solar backscatter ultraviolet radiometer 2 (SBUV/2) ozone profile and total ozone data, selected to match the UV stations' locations. The data are available from April 1970 to the present, with nearly continuous data coverage from November 1978. The satellite ozone data coverage is from backscatter ultraviolet radiometer (BUV) to solar backscatter ultraviolet radiometer 2 (SBUV-2; Bhartia et al., 2013), as follows: Nimbus-4 BUV (05/1970–04/1976), Nimbus-7 SBUV (11/1978–05/1990), NOAA-9 SBUV/2 (02/1985–01/1998), NOAA-11 SBUV/2 (01/1989–03/2001), NOAA-14 SBUV/2 (03/1995–09/2006), NOAA-16 SBUV/2 (10/2000–05/2014), NOAA-17 SBUV/2 (08/2002–03/2013), NOAA-18 SBUV/2 (07/2005–11/2012), NOAA-19 SBUV/2 (03/2009–present) and Suomi NPP OMPS (12/2011–present). We calculated daily averages by averaging the measurements from all available SBUV instruments, and then we calculated monthly means from daily averages according to Zerefos et al. (2018).

Cloud fraction monthly mean data were taken from the MODIS/Terra v6.1 dataset for the period 2000–2020. We include estimates of the variability in cloudiness around each of the ground-based monitoring stations listed in Table 1. The cloud data were taken at a spatial resolution of 3° × 3° around each monitoring station. We have calculated the correlation coefficients between the de-seasonalized monthly time series of cloud fraction from MODIS/Terra and EMAC CCM for their common period (03/2000–07/2018), in order to evaluate the model simulations. The seasonal component was removed from the series by subtracting from each monthly value the 2000–2018 seasonal mean. Analytical estimates are provided in Section 3.1 and in the Supplementary materials.

## 2.3. EMAC Chemistry climate model (CCM) simulations

We use the same CCM simulations and methodology as described by Eleftheratos et al. (2020). The simulations come from the European Centre for Medium-Range Weather Forecasts – Hamburg (ECHAM) / Modular Earth Submodel System (MESSy) Atmospheric Chemistry (EMAC) model. The EMAC model is designed to study the chemistry and dynamics of the atmosphere (Jöckel et al., 2016). The resolution applied here is 2.8$^{\circ}$ x 2.8$^{\circ}$ in latitude and longitude, with 90 model levels reaching up to 0.01 hPa (about 80 km).

We have analyzed the EMAC RC1SD-base-10 (Jöckel et al., 2016) and SC1SD-base-02 simulation results of ozone, DNA active irradiance, and total cloud cover (in %). These simulations have been performed in a "specified dynamics" (SD) setup, i.e., nudged with ECMWF ERA-Interim reanalysis data (Dee et al., 2011) for the periods January 1979 – December 2013 (RC1SD-base-10) and January 2000 – July 2018 (SC1SD-base-02), respectively, and are therefore particularly suited for a direct comparison with observations such as ground-based and satellite measurements as presented in Section 3.1 and Appendix A. We note that the SD simulation RC1SD-base-10 (which starts in 1979) is used for the comparisons during the period of ozone depletion as the SC1SD-base-02 doesn't cover that period (the 80's and 90's).

Two free running hind-case and projection simulations have been analyzed, both based on boundary conditions following the RCP-6.0 scenario: the reference simulation RC2-base-04 (1960–2100, with additional 10 years spin-up; Jöckel et al., 2016) and the sensitivity simulation SC2-fGHG-01 (1960–2100), in which the GHG mixing ratios have been kept on 1960 levels (Dhomse et al., 2018). The RC2-base-04 and SC2-fGHG-01 simulations were forced with sea surface temperatures (SSTs) and sea ice concentrations (SICs) from the Hadley Centre Global Environment Model version 2 – Earth System (HadGEM2-ES) Model (Collins et al., 2011; Martin et al., 2011). These simulations were performed for the Coupled Model Intercomparison Project – Phase 5 (CMIP5) multi-model data sets in the frame of the Program for Climate Model Diagnosis and Intercomparison (PCMDI). For years up to 2005, the data of the "historical" simulation with HadGEM2-ES is used. Afterwards, the RCP-6.0 simulation, which is initialized with the historical simulation, has been employed (Jöckel et al., 2016, and reference therein). The future solar forcing, used for the projections, has been prepared according to the solar forcing used for CMIP5 simulation of HadGEM2-ES, where the SSTs and SICs are taken from Jones et al. (2011; see also Sect. 3.3 of Jöckel et al., 2016). It consists of repetitions of an idealized solar cycle connected to the observed time series in July 2008. This has been applied consistently for all projections with prescribed SSTs (RC2-base) and the same holds also for the SC-simulations. Here, we deviate from the CCMI recommendations consisting of a sequence of the last four solar cycles (20–23) (see Sect. 3.4 of Jöckel et al., 2016).

The UV-B radiation calculated by the photolysis scheme (JVAL) (Sander et al., 2014) is weighted with the DNA damage potential of Setlow (1974) with the parameterization by Brühl and Crutzen (1989). The DNA damaging irradiance of the NDACC database is again based on the action spectrum by Setlow (1974) and parameterized using Eq. (2) of Bernhard et al. (1997). The different parameterization of the DNA action spectrum in the EMAC CCM simulations and the GB measurements will likely lead to small difference between the two datasets. For example, the radiative amplification factors (RAFs) for the two parameterizations may not be identical, which may lead to seasonal variations because RAFs are solar zenith angle and ozone dependent. To reduce such differences, we only compared de-seasonalized data. The seasonal component at each station was removed by subtracting the long-term monthly mean (2000–2018) from each individual monthly value. The monthly departures were then expressed in percent of the long-term monthly mean.

Ozone and total cloud cover data from the two RC2-base-04 and SC2-fGHG-01 free running simulations for the stations listed in Table 1 have been analyzed as well and respective de-seasonalized monthly means were derived. Here, the monthly data were de-seasonalized with respect to the 30-year long-term monthly mean (1990–2019).

We note here that by a separate analysis (not shown) on total cloud cover variability and trends through the 21[st] century, using the available simulations from the CCMI-1 REF-C2 set (e.g., Eyring et al., 2013), the EMAC models results fall well within the range of uncertainty, close to the ensemble average.

In all simulations analyzed here, we used prescribed aerosol distributions. The prescribed aerosol effects are separated into the aerosol surface area, representing chemical effects via heterogeneous chemistry, and the radiative properties influencing the radiation budget (Sect. 3.7 of Jöckel et al., 2016). Due to a glitch, the stratospheric volcanic aerosol was not considered correctly in the free running simulations (Sect. 3.12.1 of Jöckel et al., 2016). Therefore, the dynamical effects of large volcanic eruptions (e.g. Mt. Pinatubo 1991; El Chichón 1982) are essentially not represented in the simulations, except for the contribution to the tropospheric temperature signal induced by the prescribed SSTs. For the specified dynamics simulations, however, this has been corrected. Since the aerosol distributions have been prescribed, there is no aerosol output for these simulations that we could look at. As such, we cannot investigate the impact of future changes in aerosol loading on the UV radiation reaching the surface.

In the following sections, to assist the reader to easily follow the figure content, we change the notation of our simulations as follows:

- SC1SD-base-02: further abbreviated as HIS-SD for historical specified dynamics
- RC2-base-04: further denoted as REF for reference, and
- SC2-fGHG-01: further denoted as FIX for fixed GHGs.

## 2.4. Statistical methods

In Section 3.1, we evaluate the variability of DNA active irradiance from the model simulations with station measurements for a nearly 20-year period (2000–2018). The comparisons were based on regression analyses between the simulated and observed DNA active irradiance monthly data after removing variations related to the seasonal cycle. The monthly data at each station were de-seasonalized by subtracting the long-term monthly mean (2000–2018) pertaining to the same calendar month.

We have calculated the Pearson's correlation coefficients, R, between the simulated and measured DNA active irradiance (Eq. 1) and tested them for statistical significance using the t–test formula for the correlation coefficient with n–2 degrees of freedom (Eq. 2) (von Storch and Zwiers, 1999):

$$R = \frac{\sum_{i=1}^{n}(x_i - \bar{x})(y_i - \bar{y})}{\sqrt{\sum_{i=1}^{n}(x_i - \bar{x})^2}\sqrt{\sum_{i=1}^{n}(y_i - \bar{y})^2}} \tag{1}$$

Where $x$ refers to the measured and $y$ to the simulated data.

$$t = R\sqrt{\frac{n-2}{1-R^2}} \tag{2}$$

The same procedure was followed for the comparisons between the simulated and satellite derived total
ozone and cloud cover datasets for the period 2000–2018, which are presented in Section 3.1.
In Section 3.3, we apply a statistical test to compare the regression slopes in DNA active irradiances before
and after the year 2050. The null hypothesis, that the two slopes are statistically equal ($H_0: b_1 = b_2$), is
tested against the alternative hypothesis that the two slopes are not statistically equal ($H_1: b_1 \neq b_2$). The
difference in slopes is tested with the statistic:

$$t = \frac{b_1 - b_2}{s_{(b_1 - b_2)}} = \frac{b_1 - b_2}{\sqrt{s_{b_1}^2 + s_{b_2}^2}} \tag{3}$$

With $n_1 + n_2 - 4$ degrees of freedom, according to Eq. (11.20) of Armitage et al. (2002). The parameter
$s_{(b_1 - b_2)}$ is the standard error of $(b_1 - b_2)$. The parameters $b_1$ and $b_2$ are the slopes before and after 2050
in each geographical zone, and $n_1$ and $n_2$ are the numbers of data before and after 2050, respectively. The
test was performed using de-seasonalized monthly values but also with the averages shown in Figures 4c,
5c, 6c, calculated from de-seasonalized data. Both ways gave similar statistical results. In Section 3.3 we
provide the results using the de-seasonalized monthly values.
The equation for the slope of the regression line, using $x$ as the time variable and $y$ as the DNA active
irradiance variable, is:

$$b = \frac{\sum(x - \bar{x})(y - \bar{y})}{\sum(x - \bar{x})^2} \tag{4}$$

The residual mean square for the first group (1960-2049), $s_{b_1}^2$, is estimated as follows:

$$s_{b_1}^2 = \frac{\sum_{(1)}(y - Y_1)^2}{n_1 - 2} = \frac{S_{yy1} - S_{xy1}^2 / S_{xx1}}{n_1 - 2} \tag{5}$$

And the corresponding mean square for the second group (2050-2099), $s_{b_2}^2$, as follows:

$$s_{b_2}^2 = \frac{\sum_{(2)}(y - Y_2)^2}{n_2 - 2} = \frac{S_{yy2} - S_{xy2}^2 / S_{xx2}}{n_2 - 2} \tag{6}$$

Here, $S_{yy}$ is the standard deviation of DNA active irradiance, $S_{xx}$ is the standard deviation of time and
$S_{xy}$ is their covariance, for the first (1960-2049) and second (2050-2099) groups, respectively.
If $|t| > t_{critical \ (n_1 + n_2 - 4)}$, then the null hypothesis, $H_0$: the slopes are equal, is rejected at the significance
level $a$, and the alternative hypothesis (the two slopes are statistically different) is accepted.

**3 Results and discussion**

**3.1 Evaluation of EMAC CCM simulations for the present**

The time series of de-seasonalized DNA active irradiance data are presented in Figure 1. The figure compares model calculations of DNA active irradiance from the HIS-SD simulation with ground-based measurements at stations described in Section 2. The upper panel refers to the average of de-seasonalized data at three stations in the northern high latitudes, the middle panel refers to the respective average of thirteen stations between 50º N and 50º S, and the lower panel to the respective average of four stations in the southern high latitudes. The comparisons refer to the period 2000–2018. We note that this is a composite dataset, obtained with the same set of stations (both in the model and in the observations). All timeseries start from the same year in the model, but not all timeseries start from the same year in the observations.

The results show that the correlations between the simulated and ground-based DNA active irradiance data are statistically significant. Figure 1, however, shows that the variability between the simulated and ground-based data can be different. To have a better indication how the two datasets are scattered, Figure 2 presents the respective scatter plots. We have added the linear regression line and the y=x line which shows how much the slope of the fit deviates from the 1:1 line. The statistical results (correlation coefficient, slope, error, t-value, p-value, and root mean square error (RMSE)) are summarized in Table 2a.

The comparison for the near-global mean shows that the correlation coefficient between the simulated and observed DNA active irradiances is +0.709, the slope of the fit is 0.521, its error is 0.035, and the RMSE is 4.423. For the northern high latitudes, the statistical results are estimated to be R = 0.518, slope = 0.657, error of slope = 0.129, RMSE = 9.543. The respective results for the southern high latitudes are R = 0.746, slope = 0.879, error of slope = 0.070, RMSE = 14.766. It appears that the slope of the regression fit for the near-global mean is small; however, the RMSE, which is the measure of the differences between the simulated and observed values, is also small. A RMSE of 0 would indicate a perfect fit to the data, something that is never achieved in practice. It also appears that the respective RMSE of the residuals (i.e., simulated minus observed values) are larger at high latitudes; this is because it was derived from larger differences of de-seasonalized data at the northern high latitudes and even larger at the southern high latitudes.

The regression analysis results between the two data sets at each station separately are presented in Supplement Table S1. We note here that the model has a resolution of 2.8° x 2.8°, which is a large area for which the point measurement has to be representative for. As such, for stations where the surrounding area is inhomogeneous, such as mountain tops (Mauna Loa, Sonnblick, Zugspitze), or in valleys (Aosta), or by

being in a town with very heterogeneous surroundings (sea, mountains, tropospheric ozone and aerosols such as Athens), the model simulations of DNA active irradiance are not expected to be fully representative of the specific UV sites. Thus, the correlation between simulated and measured DNA weighted UV irradiance is not very good at some stations, as shown in the figures provided in the supplementary material. For the same reason, the slope between two datasets can deviate significantly from unity (see Supplement Table S1). Therefore, the comparisons at the individual stations provide a qualitative evaluation of the model's variability, but cannot be considered as a strict validation of the model. We provide here indicative estimates for individual stations, which give very good to excellent correlations: a) Summit, Greenland: R = +0.709, slope = 0.757, error of slope = 0.081, p-value <0.0001, N = 88, b) Hoher Sonnblick, Austria: R = +0.673, slope = 0.946, error of slope = 0.075, p-value <0.0001, N = 192, c) Boulder, CO, USA: R = +0.748, slope = 0.677, error of slope = 0.047, p-value <0.0001, N = 163, d) Arrival Heights, Antarctica: R = +0.939, slope = 1.000, error of slope = 0.033, p-value <0.0001, N = 126.

The same procedure was followed to evaluate simulated ozone and cloud cover. Figure 3 shows (a) ozone calculations from the HIS-SD simulation compared to satellite SBUV retrievals and (b) shows simulated cloud cover compared to cloud cover from MODIS/Terra. It appears that the variability of ozone from the model simulation follows exceptionally well the variability of ozone from the satellite retrievals. It also appears that the variability of cloud cover from the model simulation is quite well correlated with the variability from the satellite observations (see Tables 2b and 2c). The respective comparisons using zonally averaged data for the northern high latitudes ($55^\circ$ – $75^\circ$ N), the near global mean ($50^\circ$ N – $50^\circ$ S) and the southern high latitudes ($55^\circ$ – $75^\circ$ S) are presented in Supplement Figure S1. The regression results (R, slope, error of slope, t-value, p-value, and RMSE) for the large-scale latitudinal averages are presented in Supplement Table S2 and they are in line with the results from the station averages.

The Supplement Table S3 presents analytically the comparisons of total ozone between the EMAC CCM calculations and SBUV satellite retrievals. The correlations between the two different data sets are statistically significant at confidence level greater than 99.9% at all stations under study. The correlation results for four indicative stations are: a) Summit, Greenland: R = +0.927, slope = 0.791, error of slope = 0.028, p-value <0.0001, N = 131, b) Hoher Sonnblick, Austria: R = +0.902, slope = 0.803, error of slope = 0.026, p-value <0.0001, N = 223, c) Boulder, CO, USA: R = +0.854, slope = 0.757, error of slope = 0.031, p-value <0.0001, N = 223, d) Arrival Heights, Antarctica: R = +0.896, slope = 0.655, error of slope = 0.029, p-value <0.0001, N = 128.

The Supplement Table S4 presents the respective comparisons for cloud cover. The cloud observations come from MODIS/Terra. The correlation results for these four stations are: a) Summit, Greenland: R= +0.196, slope = 0.069, error of slope = 0.031, p-value = 0.025, N = 131, b) Hoher Sonnblick, Austria: R = +0.556, slope = 0.619, error of slope = 0.062, p-value <0.0001, N = 222, c) Boulder, CO, USA: R = +0.539, slope = 0.482, error of slope = 0.051, p-value <0.0001, N = 222, d) Arrival Heights, Antarctica: R = +0.537, slope = 0.949, error of slope = 0.133, p-value <0.0001, N = 129.

## 3.2 Future changes in ozone and DNA active irradiance

In the previous section we evaluated the SD simulation SC1SD-base-02 with satellite and ground-based measurements. In this section we use the EMAC CCM simulations to investigate the evolution of DNA active irradiance and the parameters that affect its long-term variability into the future. More specifically, we have analyzed the free-running simulation of the EMAC CCM, namely RC2-base-04, with increasing GHGs according to RCP-6.0 at the stations under study. An evaluation of the free running simulation RC2-base-04 with the SD simulation SC1SD-base-02 is provided in Appendix A. It helps to evaluate the quality of the results of the free running model system with respect to the SD simulation and the observations of the stations, and it serves as a "bridge" from the observations via the SD simulation results to the results of the (longer-term) free-running model simulation.

We followed the same methodology as Eleftheratos et al. (2020), to examine the effect of increasing GHGs on the evolution of DNA active radiation. We have compared the free-running simulation RC2-base-04 with the sensitivity simulation SC2-fGHG-01 where GHGs are kept constant at 1960 levels (see also Appendix A). The difference between the two free-running simulations gives us an estimate of the desired result.

We have prepared a series of figures to demonstrate the two different simulations and the differences between them. Figure 4 is based on 13 UV stations between $50^\circ$ north and south. Figure 5 shows the results for the northern high latitude stations and Figure 6 for the southern high latitude stations. The top panel refers to the evolution of total ozone anomalies from 1960 to 2100; the middle panel refers to the evolution of DNA active irradiance and the lower panel to the evolution of clouds for the same period. The left panel shows the two simulations, i.e., the free-running simulation with increasing GHGs (REF) versus the same simulation with fixed GHGs at 1960 levels (FIX) and the right panel shows their respective differences. Shown are annual averages calculated from monthly de-seasonalized data. The calculation of annual averages was done as follows: First, we de-seasonalized the monthly data at each station by subtracting the long-term monthly mean (1990–2019) pertaining to the same calendar month. Next, we calculated a monthly de-seasonalized time series for each geographical zone by averaging the monthly de-seasonalized data of the stations belonging to each geographical zone. The latter time series was used to estimate the annual data anomalies. For the northern high latitude stations, the annual average refers to the average of monthly anomalies from March to September, and for the southern high latitude stations, it refers to the average of monthly anomalies from September to March. For the stations between $50^\circ$ N–$50^\circ$ S we used all months to calculate the annual average.

In addition, we have added with green squares the DNA-weighted UV irradiance anomalies averaged at the ground-based stations under study around local noon. We also include the total ozone anomalies from SBUV with blue dots and the respective cloud cover anomalies from MODIS/Terra (magenta triangles) averaged at the stations studied. The observational data have been added to show simply that the dispersion of the simulated data matches the dispersion of the measured data.

In the study by Eleftheratos et al. (2020) data from 5 stations between 50 degrees north and south were analyzed. Here, we examine for this latitude band 13 stations instead of 5 (Figure 4). The new findings paint the same picture: an increasing trend in DNA active irradiance after the year 2050, associated with a decreasing trend in cloud cover due to the evolution of GHGs and a negative trend in total ozone (Figure 4c). Thus, our new results, based on 13 instead of 5 stations, confirm qualitatively the results of the previous study for 50º N–50º S. An offset between total ozone from SBUV and the free running simulation is evident in the 1980s, which is larger at 50º N–50º S. This is discussed later.

The focus now is at higher latitudes, which show a different picture than that of 50º N–50º S after the year 2050. At the northern high latitude stations (Figure 5), DNA active irradiance (during the summer half year) shows a decreasing trend after 2050, total ozone shows an increasing trend after 2050 and cloud cover does not show any obvious statistically significant trend. The estimated trends (in % per decade) and their standard errors are presented in Table 3. More specifically, we estimate that total ozone will increase by 1.8 ± 0.8% from 2050 to 2100 (t-value = 2.169, p-value = 0.035), DNA active irradiance will decrease by 8.2 ± 3.8% (t-value = −2.161, p-value = 0.036), and cloud cover will slightly increase by 1.4 ± 1.3% (t-value = 1.061, p-value = 0.294). Accordingly, at the southern high latitude stations (Figure 6), total ozone is estimated to increase by 4.2 ± 2.1% from 2050 to 2100 (t-value = 2.020, p-value = 0.049), DNA active irradiance is estimated to decrease by 4.8 ± 2.9% (t-value = −1.660, p-value = 0.103), and cloud cover will decrease insignificantly by 1.1 ± 1.7% (t-value = −0.604, p-value = 0.548).

The above estimates point to an increase in total ozone in the northern high latitudes by the end of the century on an almost year-round basis. In a recent study by von der Gathen et al. (2021), it was concluded that conditions favorable for large Arctic ozone loss during cold winters could persist or even worsen until the end of this century, if future abundances of GHGs continue to rise. As such, anthropogenic climate change has the potential to partially counteract the positive effects of the Montreal Protocol in protecting the Arctic ozone layer (von der Gathen et al., 2021). We examined the EMAC CCM projections regarding this finding. We have analyzed the REF and FIX simulation results of ozone, DNA active irradiance, and cloud cover for January, February, and March for the three northern high latitude stations, Summit, Barrow and Sodankylä. The trend results are presented in Table 4, which shows the trends from the two simulations, and their differences, for the periods 1960–1999, 2000–2049 and 2050–2099.

It appears that in January and February, considered as the two coldest months of the year, the trends decrease from the first (2000–2049) to the second period (2050–2099), while in March (less cold month) the picture is different. More specifically, in January, the significant positive trend of 1.53 ± 0.64% per decade in 2000–2049 changes to 0.21 ± 0.73% per decade in 2050–2099. In February, the significant positive trend of 1.79 ± 0.78% per decade in 2000–2049 decreases to 0.58 ± 0.71% per decade in 2050–2099. On the other hand, the trends in March are 0.17 ± 0.58% per decade in 2000–2049 and 1.20 ± 0.51% per decade in 2050–2099, and they agree with the general course of trends seen in Figure 5. We end up to findings that are qualitatively in agreement with those concluded by von der Gathen et al. (2021) about the

large seasonal losses of Arctic ozone during cold winters until the end of the century. We also attempted to estimate the trends in DNA active irradiance in the northern high latitude stations for January, February, and March. The results are presented in Table 4 for the two periods, 2000–2049 and 2050–2099, but due to the polar night at the northern high latitudes, UV values are very low in January and February, and the predicted trends have large standard errors. As such, they are not analyzed any further.

Another issue is that Figure 5a suggests that clouds will stay more or less constant over the Arctic. Other models predict that cloud cover in the Arctic will increase until the end of the century. With sea ice diminishing in the Arctic, evaporation would increase, leading to more moisture in the air, resulting in more clouds, which in turn is expected to reduce UV radiation. For example, Fountoulakis and Bais (2015) analyzed changes in UV radiation projected for the Arctic. Comparison of Figure 1 (clear-sky trends) and Figure 2 (all-sky trends) of Fountoulakis and Bais (2015) suggests that UV changes between the future and the present will become more negative when clouds are also considered due to the projected increase in cloud attenuation. Our estimates indicate a significant cloud increase of ~2.2% from 1960 to 2100 (~1.4% from 2050 to 2100, not significant). These increases are small and are based on the average of three stations only, Summit, Barrow and Sodankylä, but they are in accordance with the results obtained for the whole latitudinal band of 55$^{\circ}$–75$^{\circ}$ N (~2.7% from 1960 to 2100, p-value < 0.0001, and ~0.9% from 2050 to 2100, p-value = 0.05). Summit and Sodankylä are far away from the seashore and are not affected by the ocean, while Barrow is located only 250 m away from the coast and is greatly affected by the ocean. Changes in cloudiness might be different at coastal and mainland sites. For Barrow (coastal site) we estimate a significant cloud increase of 5.5% in the period 1960–2100 (3% in the period 2050–2099), while for Summit (pure land site) we estimate an insignificant change of –0.1% in the period 1960–2100 (–0.4% in the period 2050–2100). For Sodankylä (also pure land site), we estimate an insignificant increase of 1.2% in the period 1960–2100 (p-value = 0.365), and of 1.6% in the period 2050–2100 (p-value = 0.446). Averaging large and small changes in cloudiness should finally result to moderate changes. These results generally agree with the results presented in other studies (Bais et al., 2015; Fountoulakis and Bais, 2015) for land areas of the Arctic (keeping also in mind that the results of the present study are averages for three stations only). We note that the results presented in these two referenced studies were for RCPs 4.5 and 8.5, and thus not directly comparable with the results of our study. In a more recent study presenting RCP 6.0-based projections (Bais et al., 2019), it was shown that cloudiness changes at high latitudes would strongly affect the UV irradiance mainly over the ocean where the absence of sea ice would result to increased evaporation. For land, smaller and non-significant changes were reported (see Figure 8 of Bais et al., 2019), which is again in agreement with the results presented in our study. In another study (Figure 5 of Fountoulakis et al., 2014), changes in zonally averaged UV irradiance due to changes in cloudiness in 1950–2100 were estimated to be the order of 5–15% (depending on the RCP) for latitudes ~70 degrees. However, only changes over the ocean were considered in that study and not over land. Additional indications that our results should be considered representative of the three stations under study and not the entire Arctic region is provided in Figure B1 (Appendix B), which shows the changes in zonal mean cloud

cover for the Arctic region from the REF simulation. It appears that the zonal mean cloudiness is expected
to increase more and more as move northward of 50° towards the North pole, indicating that the largest
changes in cloud cover are likely to occur over the ocean and not over land.
For the period 1960–1999, the DNA active irradiance (summer half year) showed upward trends in all
geographical zones following the downward trends of total ozone. Nevertheless, we should note that the
examined simulation REF (simulation with full chemistry and increasing GHGs according to RCP-6.0)
seems to clearly underestimate the observed ozone depletion of the 1980s and 1990s in the geographical
region 50° N–50° S (Figure 4), but in the higher latitude regions (Figs. 5 and 6) the picture looks much
better. This suggests that there may be a bias in the model, that might at least partly be caused by not
considering all ozone depleting substances (ODSs), but only a subset (only CFC-11 and CFC-12 are
considered; Jöckel et al., 2016). The REF simulation also underestimates the ozone depletion of the 1980s
and 1990s in the northern high latitude stations (Figure 5), but the picture is better than that of 50° N–50° S.
The FIX simulation seems to reproduce better the Arctic ozone depletion of the past. The latter, however, is
coincidence; it only indicates that due to the higher dynamic variability of the northern (winter)
stratosphere, the evolution of the ozone layer in the Arctic region is significantly affected by natural
variability of the stratosphere due to planetary waves. The best agreement between the REF simulation and
satellite measurements during the period of ozone depletion is found for the southern high latitudes, as can
be seen from Figure 6. As such, we can infer that the model simulations reproduce very well the observed
ozone depletion of the past in particular in the southern higher latitudes, and less well in the northern higher
latitudes. Nevertheless, the simulated decline of ozone during 1979–1999 and the minimum ozone values
calculated by the model in the 1990s for the near global mean (50° N-50° S) and for the higher latitudes,
are qualitatively in line with the satellite ozone observations, which is a good outcome. This is supported by
Figure A1 (Appendix A), which shows the free running simulation REF against the SD simulation HIS-SD
which starts in 2000. Because we wanted to evaluate the free running simulation for the period of ozone
depletion, we also analyzed the SD simulation RC1SD-base-10 which starts in 1979. It appears that the
REF simulation seems to reproduce well the negative ozone trends during the period of ozone depletion,
but not the exact anomalies of a particular year. This is because the free running simulation has its own
meteorological/synoptical sequence, and thus we cannot expect that the observed time series of the past is
reproduced on a year-by-year basis in the free running simulation the same way is reproduced in the
simulation with "specified dynamics".
Finally, we should also refer to the recent assessment of the United Nations Environment Programme
(UNEP) Environmental Assessment Panel (EEAP) (Bernhard et al., 2020), which compared projections of
future UV radiation from two studies, Bais et al. (2019) and Lamy et al. (2019). We have compared our
trend estimates, which are based on one model only, with the estimates provided in Table 1 of Bernhard et
al. (2020), which are based on many models of the first phase of the Chemistry-Climate Model Initiative
(CCMI-1) and should therefore be considered more robust than the estimates provided here. We clarify that
it is only a qualitative comparison as our trends are based on DNA weighted irradiance while the table in
Bernhard et al. (2020) refers to erythema. The DNA radiation amplification factor is about 2.1 while that
for erythema is 1.2, which suggests that we would expect differences in trends by roughly a factor of 1.75.
We also note that the table of Bernhard et al. (2020) shows zonal mean changes of the clear-sky UV index,
whereas we estimate changes in DNA active irradiance based on station averages. Despite the
inconsistencies in the radiation fields being compared, our trend estimates from the REF simulation based
on RCP-6.0 are qualitatively in line with the results presented by Bernhard et al. (2020) for the case of
RPC-6.0. We estimate a statistically significant decrease in DNA active irradiance at the northern high
latitude stations for the period 2015 to 2090 of about –17% (–16% for the zonal mean 55°–75° N). The
numbers from Table 1 of Bernhard et al. (2020) for the northern high latitudes are –6% for the annual mean
clear-sky UV index for the period 2015 to 2090, and –3%, –7%, –5% and –4% for January, April, July, and
October, respectively. Our respective estimate for the southern high latitude stations is about –24% (–26%
for the zonal mean 55°–75° S) and is also qualitatively in line with the negative trend estimates provided by
Bernhard et al. (2020) for the southern high latitudes for the period 2015 to 2090 (–18% for the annual
mean clear-sky UV index, and –8%, –6%, –6% and –23% for January, April, July, and October,
respectively).
The above estimates are based on station averages. We have complemented the analysis presented in
Figures 4, 5, 6 with zonally averaged data in order not to restrict the analysis of the model results to only 20
locations but to analyse the model results as a whole, for example in latitudinal bands. The Supplement
Figures S2, S3, S4 show the changes from the free running simulations REF, FIX, and their differences, for
the near global mean, the northern and southern high latitudes based on latitudinal averages. It appears that
the results from the analysis of averaging the model data in latitudinal bands are in the same direction with
that from the station averages. More specifically, for the near global mean we find similar results to those
presented in Figure 4 for the station mean, but a stronger negative trend in total ozone after 2050 which
together with the negative cloud trend drive the positive DNA active radiation trend after 2050. On the
other hand, negative trends in ozone and clouds after 2050 are not observed in the northern or the southern
high latitude belt.
We believe that the negative trends seen in the near-global mean after 2050 will result from the ongoing
climate change that will affect ozone and clouds. It is well known that increasing GHG concentrations have
led to tropospheric warming and stratospheric cooling over the last decades (Stocker et al., 2013; Zerefos et
al., 2014). As a thermodynamic consequence, the troposphere has expanded and the height of the
tropopause has increased (Santer et al., 2003). A recent study showed that the stratospheric layer has
contracted substantially over the last decades due to increasing GHGs and will continue to contract under
the RCP-6.0 scenario (Pisoft et al., 2021). Also, chemistry-climate and climate model projection results
show an acceleration in Brewer-Dobson circulation in response to climate change (Butchard, 2014). These
changes will not leave the ozone layer unaffected. Our model simulations for the near-global mean show a
downward trend in total ozone after 2050, which will likely be shaped by the negative trends in the lower
stratosphere due increasing GHGs. For clouds, it has been shown that increasing GHGs are responsible for
less clouds over the mid-latitudes (Norris et al., 2016) and for breaking up stratocumulus clouds into
scattered clouds (Schneider et al., 2019). Our simulations show that clouds will decrease in response to
increasing GHGs, which is consistent with the indications from these studies.
The results based on the zonally averaged data fully support the basic finding of the paper that DNA active
irradiance is expected to change differently at high latitudes than at near-global scale after around 2050. It
will continue to decline at high latitudes mainly due to ozone recovery from the reduction of ODSs (cloud
cover changes are not significant), while it is expected to increase on a near-global scale, affected by
reductions in cloud cover and total ozone due to climate change. This of course is an outcome that emerges
from the simulations of a single climate-chemistry model, and as such, it may well turn out to be true or
false. Verification of the results from other model simulations would be important, but also important is the
further investigation of cloud behaviour from the model simulations and their verification from high quality
measurements. It is important to note that our free running simulations were designed according to the
definitions for the reference and sensitivity simulations provided by the IGAC and SPARC communities to
address emerging science questions, improve process understanding and support upcoming ozone and
climate assessments (Eyring et al., 2013).

**3.3 Statistical evaluation of differences between trends and statistical modelling**
We have compared the regression slopes in DNA active irradiances before and after the year 2050.
Our calculations according to Eq. 3 show that at the significance level $\alpha = 0.05$, the null hypothesis that the
slopes are statistically equal, cannot be rejected for neither the northern, nor the southern high latitudes
(>55º), and therefore we cannot conclude that there is any statistically significant difference between the
trends in DNA active irradiance before and after 2050 in these two latitude zones. On the other hand, the
null hypothesis is rejected for the latitude zone of 50º N–50º S, which means that the alternative hypothesis
is accepted, and so the two trends before and after 2050 are statistically different. The statistical results are
presented in Table 5.
We note here that the statistical test was also applied for the periods before 2050, i.e., the two periods 1960-
1999 and 2000-2049, to test if their trends are statistically significant or not. In all latitudes it was found
that the regression slope of the period 1960-1999 is not statistically significantly different from the
regression slope of the period 2000-2049. As such, it appears that only after the year 2050 there appears to
be a statistically significant change in the trends of DNA active irradiance because of the evolution of
GHGs and only at latitudes between 50º N and 50º S. At latitudes poleward of 55º, the DNA active
irradiance is more likely to continue to decrease due to the increasing ozone trends from the reduction of
the concentrations of ODSs.
Moreover, we have applied multiple linear regression (MLR) analysis to examine the contribution of ozone
and cloud trends to the estimated DNA active irradiance trends after the year 2050. The MLR model was
applied to the differences between the two model simulations, REF and FIX, which were estimated from
monthly de-seasonalized data (*deseas*). The MLR model is of the following form:

$$deseas\ DNA\ active\ irradiance = a + \beta_{O_3} \cdot deseas\ O_3 + \beta_{cloud} \cdot deseas\ Cloud \quad (7)$$

Where, $a$ is the intercept, $\beta_{O_3}$ is the ozone coefficient and $\beta_{cloud}$ is the cloud coefficient for the period
2050–2099. The regression coefficients and their standard errors are presented in Table 6a. These
coefficients were derived from station mean data, and hence might not be representative for the entire
geographical zones. As can be seen, the coefficients $\beta_{O_3}$ and $\beta_{cloud}$ are highly statistically significant with
small errors in all cases (p-values <0.001). We have used the regression coefficients to determine the part
of the DNA active irradiance trends that are caused by trends in total ozone and cloud cover. We have
derived the ozone-related DNA active irradiance trend by multiplying the regression coefficient between
DNA active irradiance and ozone ($\beta_{O_3}$) with the trend in ozone for the period 2050–2099. Accordingly, we
derived the respective cloud-related DNA active irradiance trend by multiplying the regression coefficient
$\beta_{cloud}$ with the cloud trend.
For the northern high latitude stations (>55º N), we estimate an ozone-related DNA active irradiance trend
of about −0.72% per decade, indicating that ~39% of the DNA active irradiance trend (−1.86% per decade)
is caused by the trends in ozone. The respective cloud-related DNA active irradiance trend is smaller
(−0.21% per decade), which means that the cloud trend explains ~11% of the DNA active irradiance trend.
Both parameters account for ~50% of the predicted DNA active irradiance trend. The remaining part of the
DNA active irradiance trend is related to changes in other parameters, as for instance in surface albedo, as
is discussed later in Section 3.4.
Similar results regarding the contribution of ozone and cloud trends to the predicted DNA active irradiance
trend are also found for the southern high latitude stations (>55º S), but not for the stations averaged
between 50º N and 50º S. The results are summarized in Table 6b. For the southern high latitude stations
(>55º S), the ozone-related DNA active irradiance trend is −0.57% per decade and the cloud-related DNA
active irradiance trend is +0.07% per decade. As such, ~59% of the DNA active irradiance trend (−0.96%
per decade) is explained by ozone, and ~7% is explained by clouds.
For stations averaged between 50º N–50º S, we estimate that the ozone-related DNA active irradiance trend
is +0.27% per decade, and the cloud-related DNA active irradiance trend is +0.33% per decade. The
contribution of changes in cloudiness is larger than the contribution of changes in ozone (~41% compared
to ~33%, respectively), and therefore, our findings support the previous results by Eleftheratos et al. (2020),
who analyzed a smaller number of GB stations between 50º N–50º S than those used here.
**3.4     Changes in surface albedo and relation to DNA active irradiance**
In the previous section we showed that DNA active irradiance will continue to decrease after the year 2050
at high latitudes as a result of ozone change rather than cloud cover change. Another parameter affecting
the solar UV variability at high latitudes is surface albedo (Weihs et al., 1999; Nichol et al., 2003;
Weatherhead et al., 2005; Gröbner, 2012; Bais et al., 2019). In this respect, changes in surface albedo are
expected to affect the long-term variability of surface UV-B irradiance. Figure 6 shows the changes in
surface albedo simulated with the EMAC CCM at the two stations, Barrow in Alaska and Palmer in
Antarctica. More specifically the figure shows the differences between the two model simulations, the one
with increasing GHGs (REF) and the one with fixed GHGs (FIX), in order to account also for the effect of
increasing GHGs on surface albedo changes according to the methodology applied in Section 3.2. The
results refer to the summer seasons of the two hemispheres, where there is sufficient sunlight in the Arctic
and the Antarctic. Table 7 summarizes the trends in the differences between the two model simulations,
REF and FIX, for the DNA active irradiance, total ozone, cloud cover and surface albedo at Barrow
(Alaska) and Palmer (Antarctica) for the periods 1960–1999, 2000–2049 and 2050–2099. While variations
in surface albedo are certainly of primary importance for high-latitude sites, they can play a non-negligible
role even at mid-latitudes. However, they were not analyzed here.
From Figure 7 it is clear that surface albedo decreases significantly by the end of the 21$^{st}$ century in view of
the increasing GHG emissions. The decreases in surface albedo (Table 7) are larger in Barrow (Alaska)
than Palmer (Antarctica). The trend for Barrow is qualitatively consistent with the conclusion by Bernhard
(2011), showing that the ground at Barrow is covered by snow later and later at the start of winter. We also
note that both, Barrow and Palmer, are coastal sites and are heavily affected by local conditions (e.g., how
far sea ice gets to the station), which may not be simulated correctly. Therefore, we point out that the
evolution of albedo at the two stations shown in Figure 7 is representative for regional changes but may not
accurately reflect changes at the exact location of these stations.
To assess the impact of the albedo changes on UV variability, we used surface albedo as additional
explanatory variable in the MLR model of Eq. (7). We determined an additional regression coefficient,
namely $\beta_{albedo}$, which explains the effect of albedo change on DNA active irradiance change at the two
stations under study, Barrow and Palmer. We estimated an albedo-related DNA active irradiance trend, in
the same way as described above, by multiplying the coefficient $\beta_{albedo}$ with the trend in albedo differences
between the two model simulations.
For Barrow, we estimate an ozone-related DNA active irradiance trend of about −0.87% per decade for the
period 2050–2099, indicating that ~41% of the DNA active irradiance trend (−2.14% per decade) is caused
by trends in ozone. The respective cloud-related DNA active irradiance trend is about −0.49% per decade,
which means that the cloud trend explains ~23% of the DNA active irradiance trend. The surface albedo-
related DNA radiation trend is about −0.45% per decade, explaining ~21% of the DNA active irradiance
trend in the period 2050-2099. The model suggests that all parameters together explain ~85% of the DNA
active irradiance trend, which however may not be such an unbiased result. This is because the effects of
clouds and albedo are not independent, as assumed in the regression equation. For 100% albedo and non-
absorbing clouds, clouds would barely attenuate UV radiation. For actual albedo and cloud conditions,
clouds do attenuate, but the effect is greatly reduced by surface albedo because of multiple reflections
between surface and cloud (Nichol et al., 2003).
At Palmer, the trends are smaller. The ozone-related DNA active irradiance trend is −0.46% per decade, the
cloud-related DNA active irradiance trend is 0.43% per decade, and the albedo-related DNA active
irradiance trend is −0.31% per decade. These trends together determine the small negative trend, which is
predicted for the DNA active UV irradiance in the period 2050–2099 of about −0.33% per decade.
The above calculations indicate that the impact of albedo trends on DNA active irradiance trends due to the
continuous increase of GHGs until the end of the 21$^{st}$ century is important and should not be ignored when
studying the long-term changes of DNA active radiation reaching the ground. The model simulations at
Barrow and Palmer suggest that the surface albedo changes might be larger at Barrow than Palmer
according to Table 7. The model simulations also suggest that the northern high latitudes might experience
larger changes in surface albedo than the southern high latitudes in the period 2050–2100 (Appendix C,
Figures C1 and C2).
In order to better represent the northern and southern high latitudes, we also applied the MLR model to the
large-scale zonal means of 55$^{o}$ – 75$^{o}$ N and 55$^{o}$ – 75$^{o}$ S. For the northern high latitude zone, the findings are
in the same direction as those found for Barrow. We estimate that ~31% of the DNA active irradiance trend
is determined by the trend in ozone, and that ~14% and ~32% of the DNA active radiation trend are
explained by trends in clouds and surface albedo, respectively. For the southern high latitude zone, we
estimate that the largest part of the DNA active irradiance trend is determined by the trend in ozone, and
that the contributions of cloud and albedo trends are small.
**4 Summary and Conclusions**
We have studied changes in ozone, DNA active irradiance and cloud cover due to the evolution of
greenhouse gas concentrations in the near global mean (50° N–50° S) and in the northern and southern high
latitudes, using the EMAC CCM simulations from 1960 to 2100.
The model simulations have been evaluated against ground-based UV irradiance measurements, satellite
ozone observations from SBUV (v8.7) and satellite cloud fraction data from MODIS/Terra for the period
2000–2018. The evaluation results can be summarized as follows:
• Simulations of total ozone with specified dynamics (RC1SD-base-10 and HIS-SD) reproduce
extremely well the variability of total ozone in the northern and southern high latitudes for the
periods 1979–2013 and 2000–2018, respectively. The correlation analysis results between EMAC
HIS-SD simulation and SBUV (v8.7) satellite ozone de-seasonalized data are: Northern high
latitudes (3 station mean), R = +0.899, p-value <0.0001; Southern high latitudes (4 station mean),
R = +0.892, p-value <0.0001; 50º N–50º S (13 station mean), R = +0.894, p-value <0.0001.
• The respective simulations of DNA active irradiance correlate quite well with ground-based UV
measurements, as follows: Northern high latitudes (3 station mean), R = +0.504, p-value <0.0001;
Southern high latitudes (4 station mean), R = +0.746, p-value <0.0001; 50º N–50º S (13 station
mean, R = +0.499, p-value <0.0001).
• Evaluation of cloud cover simulations against MODIS/Terra cloud fraction data gave good
correlations as follows: Northern high latitudes (3 stations mean), R = +0.453, p-value <0.0001;
Southern high latitudes (4 station mean), R = +0.485, p-value <0.0001; 50º N–50º S, R = +0.703,
p-value <0.0001.
Between 50º N–50º S, the DNA-damaging UV radiation is expected to decrease until 2050 and to increase
thereafter. This increase is associated with expected decreases in cloud cover and insignificant trends in
total ozone, as it was shown previously by Eleftheratos et al. (2020). Our study however expands the
previous work by adding more stations in low and mid-latitudes and by including estimates from high
latitude stations with long-term measurements of UV irradiance.
In contrast to the predictions for 50º N–50º S, we estimate that DNA active irradiance will continue to
decrease after the year 2050 in the northern and southern high latitudes (>55º) due to increasing ozone.
More specifically, for the northern high latitude stations we estimate that total ozone will increase by 1.8 ±
0.8% from 2050 to 2100, DNA active irradiance will decrease by 8.2 ± 3.8% and that cloud cover will
increase insignificantly by 1.4 ± 1.3%. Similarly, in the southern high latitude stations, total ozone is
estimated to increase by 4.2 ± 2.1% from 2050 to 2100, DNA active irradiance is estimated to decrease by
4.8 ± 2.9% and cloud cover will decrease insignificantly by 1.1 ± 1.7%.
The statistical results have been confirmed by statistical tests. Statistical comparisons of the regression
slopes before and after 2050 in the northern and southern high latitude stations under study showed that
there are no statistically significant different trends in DNA active irradiance before and after that year. On
the other hand, between 50º N–50º S the trends before and after 2050 were found to be statistically
significantly different at the 0.05 significance level. The test confirmed the statistical result that DNA
active irradiance will reverse sign and become positive after 2050 at stations between 50º N–50º S mainly
due to cloud cover and ozone changes associated with climate change, something that is likely not to
happen at high latitudes, where the DNA-damaging UV-B radiation is projected to continue its downward
trend after 2050 mainly due to the continued increase of ozone from the reduction of ODSs. In addition, it
should be mentioned, that the enhanced GHG concentrations will cool the stratosphere and therefore the
stratospheric ozone content (especially in the middle and upper stratosphere) is expected to increase
because the ozone depleting reactions (homogeneous gas phase reactions) will be getting slower. From
Dhomse et al. (2018) we know that the (future) Arctic and the Antarctic stratosphere are developing
differently in spring. In particular, the Arctic region is indicating a stronger reaction on enhanced GHG
concentrations (most likely due to the dynamic feedbacks in the northern hemisphere, i.e., related to the
planetary wave activity).
We clarify here that our findings for the high latitudes refer to the summer periods and not to the seasons
when ozone depletion occurs, for which it has been shown that climate change will favor large spring loss
of Arctic column ozone in connection with extraordinary (persistent) cold stratospheric winters (with low
planetary wave activity) in the future (von der Gathen et al., 2021). The best agreement between the REF
simulation results and satellite measurements during the period of ozone depletion was found for the
southern high latitudes. The REF simulation (full chemistry and increasing GHGs according to RCP-6.0)
seems to underestimate the observed ozone depletion of the 1980s and 1990s for the near global mean (50$^\circ$
N–50$^\circ$ S) and at high latitudes of the Northern Hemisphere. This might at least partly be caused by not
considering all ODSs, but only a subset (only CFC-11 and CFC-12 were considered). Despite this feature,
the simulated ozone declines during 1979–1999 and the minimum ozone values calculated by the model in
the 1990s for the northern mid- and high latitudes, are qualitatively in line with the satellite ozone
observations.
Also, our analysis suggests that clouds might stay constant over the Arctic, while other models predict that
cloud cover in the Arctic will increase during the next decades due to enhanced evaporation of water vapor
by the sea-ice decrease. Our estimates, however, refer to two sites in the Arctic and not to the entire Arctic
Ocean. As such, our results should be considered representative of the land sites under study and not of the
entire Arctic or Antarctic regions. In addition, we cannot reliably evaluate the projection of cloud cover
over time, using MODIS observations for a relatively short period. So, in the end we must trust that the
physics coded in the model is correct. Hence, verification of our results using independent CCMs would be
highly desired. We conducted a separate analysis on total cloud cover variability and trends through the 21$^{st}$
century, using the available simulations from the CCMI-1 REF-C2 set, which showed that the EMAC CCM
results fall well within the range of uncertainty (i.e., $\pm 2\sigma$), close to the ensemble average ($\pm 1\sigma$).
Moreover, we applied a multiple linear regression model to examine the contribution of ozone and cloud
trends to the estimated DNA active irradiance trends after the year 2050. The model was applied to the
differences between the two model simulations, REF and FIX. It was found that ozone is the primary
contributor accounting for about ~50% of the predicted trends in DNA active irradiance after 2050 both in
the northern and in the southern high latitude stations.
The impact of surface albedo on DNA active irradiance trends due to the evolution of GHGs (RCP-6.0) has
been examined at two stations, Barrow in the Arctic, and Palmer in the Antarctic. The model simulations
suggest that declining trends in surface albedo are larger at Barrow than Palmer. The driving force for the
decrease in Arctic surface albedo is by 70% the decrease in snow cover fraction over the Arctic land and
sea-ice due to the increase in surface air temperature and decrease in snowfall (Zhang et al., 2019).
Unlike the Arctic sea-ice, which has consistently declined over the past four decades, the Antarctic sea-ice
has shown little change (increase) from 1979 to 2015 but large regional and temporal variability (Maksym,
2019). A rapid decline in 2015–2018, far exceeding the decreasing rates seen in the Arctic (Parkinson,
2019), may have foreboded future changes in Antarctic sea-ice (Eayrs et al., 2021). The observed decline
lowered the region's surface albedo, highlighting the importance of Antarctic sea-ice loss to the global
snow and ice albedo feedback (Riihelä et al., 2021). This sea-ice reduction probably resulted from the
interaction of a decades-long ocean warming trend and an early spring southward advection of atmospheric
heat, with an exceptional weakening of the Southern Hemisphere mid-latitude westerlies in late spring
(Eayrs et al., 2021). Obviously, such abrupt declines cannot be predicted by the present-day model
simulations. This is because the mechanisms for the Antarctic sea-ice variations are not yet well understood
and future predictions are highly uncertain.
IPCC (2021) concluded that there has been no significant trend in Antarctic sea-ice area from 1979 to 2020,
due to regionally opposing trends and large internal variability. In the Bellingshausen and Amundsen Seas,
however, the observed sea-ice has shown decreasing trends (Maksym, 2019; Parkinson, 2019; Eayrs et al.,
2021). Our estimates for Palmer, which is located at the coast of the Bellingshausen Sea, shows a negative
trend in surface albedo from 1979 to 2020, which is in line with the negative trends in sea-ice observed in
Bellingshausen and Amundsen Seas. The REF simulation shows that the surface albedo at Palmer will
continue to decrease until 2100. This result should be considered representative of the Palmer station and
its surroundings, and not of the entire Antarctic region.
The key findings presented in this study are that model and measurements agree fairly well, giving support
to the simulations of the future scenarios. Cloud cover is generally decreasing, leading to increased solar
radiation, apart from the high latitudes, where no significant changes are observed. Total ozone shows an
increasing trend from the reduction of ODSs, while a decrease is observed after 2050 on a near-global scale
due to the increase of GHGs. UV trends are a combination of changes in ozone and cloud cover, while at
high latitudes, decreased surface albedo in the second half of the century has a significant influence on the
surface UV radiation.
The above findings were based on the analysis of model simulations from 3 stations in the northern high
latitudes, 4 stations in the southern high latitudes, and 13 stations in the near-global mean with
contributions mainly from the mid-latitudes. A separate analysis using zonal means showed that the results
from the analysis of the model data averaged over geographical zones are qualitatively in the same
direction with those from the station averages. All simulations were based on a single CCM, and therefore,
verification of the results from simulations of other models would be quite useful.
**Appendix A Qualitative evaluation of free running CCM simulations against simulations with**
**specified dynamics**
In this appendix, we compare the free running ozone simulation REF, with the SD simulation RC1SD-base-
10 and SBUV satellite ozone data (v8.7). The simulation with specific dynamics RC1SD-base-10 covers
the period January 1979 – December 2013. The simulation has been used in recent assessments reports for
stratospheric ozone studies (e.g., LOTUS, 2019). In addition to the nudging towards ECMWF ERA-Interim
(Dee at al., 2011) reanalysis data (for details about the nudging setup see Jöckel et al., 2016) the simulation
uses also sea surface temperatures and sea-ice concentrations from the ERA Interim reanalysis data. Here,
we use the SD simulation (RC1SD-base-10) to show that the free running simulation (REF) is capable to
qualitatively reflect the negative ozone trends of the 1980s and 1990s. The reason for quoting the RC1SD-
base-10 simulation is because the HIS-SD simulation that is used in section 3.1 does not go back in time
before 2000, and therefore we cannot qualitatively evaluate our free running simulation before 2000. We
also appose here the SD simulation (HIS-SD), which covers the period January 2000 – July 2018. This is
useful and helpful to classify the results of the free running model system concerning the quality with
respect to the SD simulation and the observations of the stations, and it serves as a "bridge" from the
observations via the SD simulation results, to the results of the (longer-term) free-running model
simulation.
Figure A1 shows the comparison between the simulations and SBUV data. Obviously, the RC1SD-base-10
simulation (period 1979–2013) compares much better with the SBUV data than the REF simulation. The
same also holds for the HIS-SD simulation (period 2000–2018). This is expected since the SD simulation
uses reanalyzed meteorology, whereas the free running simulation has its own meteorological/synoptical
sequence. For comparison with the fixed GHG simulation, we need to switch to the pair of free running
simulations. And the question is, if the evaluation (comparison with observations) also hold for the REF
simulation, which is the basis for the comparison with the fixed GHG simulation (FIX). In the case of free
running simulations, the evaluation is only possible for the trends and for the amplitude of the year-to-year
variability, but not for the sign of the anomaly in a given nominal year and/or month. Figure A1 shows that
the free running simulation (REF) reflects correctly the negative ozone trends of the past, seen in the
observations and in the SD simulation, and is therefore suitable for comparison with the fixed GHG
simulation.

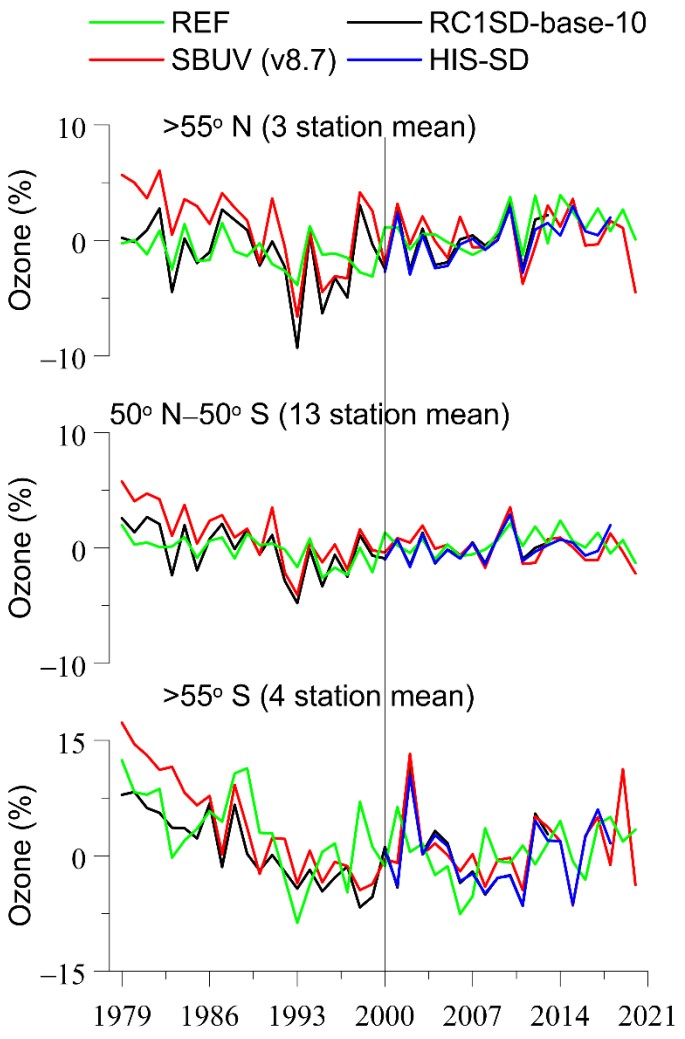


**Figure A1.** Comparison of REF (free running simulation; green line) with RC1SD-base-10 (SD simulation for 1979–2013; black line), HIS-SD (SD simulation for 2000–2018; blue line) and SBUV (v8.7) satellite measurements (red line) for 3 stations higher than 55º N (upper plot), 13 stations between 50º N–50º S (middle plot) and 4 stations higher than 55º S (lower plot). The vertical line has been put in the year 2000. The y-axis shows yearly averaged total ozone data (in %) calculated from de-seasonalized monthly data. The monthly data were de-seasonalized relative to the long-term monthly mean (2000–2018) and were expressed in %. For the northern high latitude stations, the annual average refers to the average of monthly anomalies from March to September, and for the southern high latitude stations, it refers to the average of monthly anomalies from September to March. For the stations between 50º N–50º S we used all months to calculate the annual average.

827

**Appendix B Model simulations of zonally averaged cloud cover between 50º and 80º N**

Figure B1 shows the changes of the zonally averaged cloud cover based on REF (RCP-6.0) and FIX simulations, and their differences (REF minus FIX), per 10-degree latitude zones from 50º to 80º N. For the period 1960 to 2100, the changes in cloud cover due to the evolution of GHGs (RCP-6.0) are presented in Table B1. The same picture with increasing trends as we move northward of 50º N is also found for the period 2050 to 2100.

**Table B1.** Changes in zonal mean cloud cover between 50º and 80º N due to the evolution of GHGs (RCP-6.0), for the periods 1960–2100 and 2050–2100.

|  | 1960–2100 | | | 2050–2100 | | |
|---|---|---|---|---|---|---|
|  | % Change | p-value | N | % Change | p-value | N |
| 50º–60º N | 0.9 | < 0.0001 | 140 | 0.3 | 0.56064 | 50 |
| 60º–70º N | 2.7 | < 0.0001 | 140 | 0.7 | 0.27113 | 50 |
| 70º–80º N | 4.3 | < 0.0001 | 140 | 1.9 | 0.00012 | 50 |

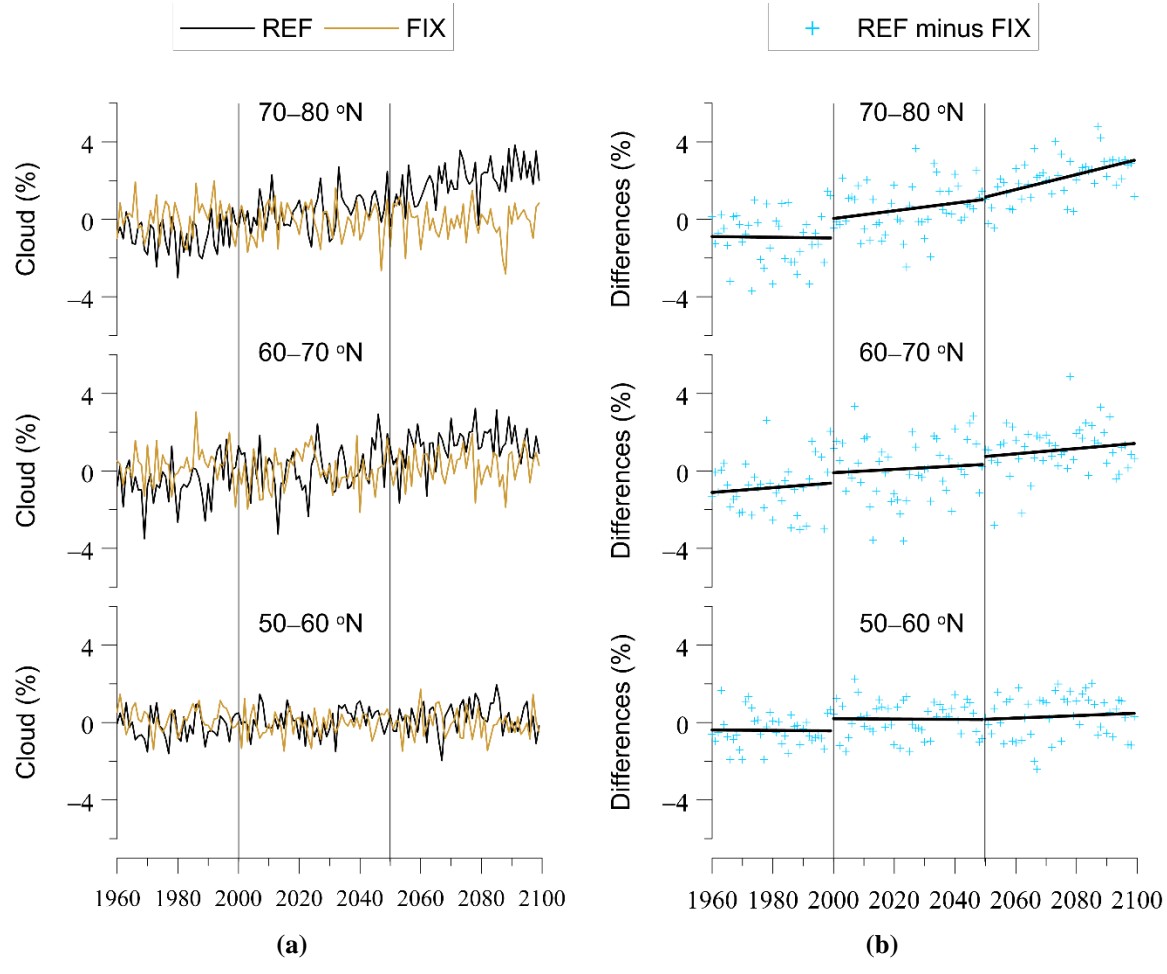

**(a)**    **(b)**

**Figure B1.** EMAC CCM projections of zonal mean cloud cover for 10-degree latitude zones (50º–60º N,
60º–70º N, 70º–80º N), based on simulations with increasing and fixed GHGs mixing ratios. **(a)** REF is the
simulation with increasing GHGs according to RCP-6.0. FIX is the simulation with fixed GHGs emissions
at 1960 levels. **(b)** Difference between the two model simulations, as an indicator of the impact of
increasing GHGs. The y-axis in the left figure (a) shows yearly averaged cloud cover data (in %) calculated
from de-seasonalized monthly data. The monthly data were de-seasonalized relative to the long-term
monthly mean (1990–2019) and were expressed in %. For the northern high latitudes, the annual average
refers to the average of monthly anomalies from March to September.

**Appendix C Model simulations of zonally averaged surface albedo between 50º and 80º N, and 50º**
**and 80º S**
Figure C1 shows the changes in zonally averaged surface albedo based on REF (RCP-6.0) and FIX
simulations, and their differences (REF minus FIX), per 10-degree latitude zones between 50º N and 80º N.
Figure C2 shows the respective changes between 50º S and 80º S. The changes in surface albedo due to the
evolution of GHGs (RCP-6.0) between 50º and 80º N, and 50º and 80º S, are summarized in Table C1.

**Table C1.** Changes in zonal mean surface albedo due to the evolution of GHGs (RCP-6.0) between 50º and
80º N, and 50º and 80º S, for the periods 1960–2100 and 2050–2100.

| North | 1960–2100 | | | 2050–2100 | | |
|---|---|---|---|---|---|---|
| | % Change | p-value | N | % Change | p-value | N |
| 50º–60º N | −21.0 | < 0.0001 | 140 | −8.9 | < 0.0001 | 50 |
| 60º–70º N | −18.3 | < 0.0001 | 140 | −9.2 | < 0.0001 | 50 |
| 70º–80º N | −41.3 | < 0.0001 | 140 | −15.1 | < 0.0001 | 50 |


| South | 1960–2100 | | | 2050–2100 | | |
|---|---|---|---|---|---|---|
| | % Change | p-value | N | % Change | p-value | N |
| 50º–60º S | −12.5 | < 0.0001 | 140 | −3.7 | 0.00299 | 50 |
| 60º–70º S | −22.5 | < 0.0001 | 140 | −3.8 | 0.00298 | 50 |
| 70º–80º S | −6.1 | < 0.0001 | 140 | −1.3 | 0.00132 | 50 |



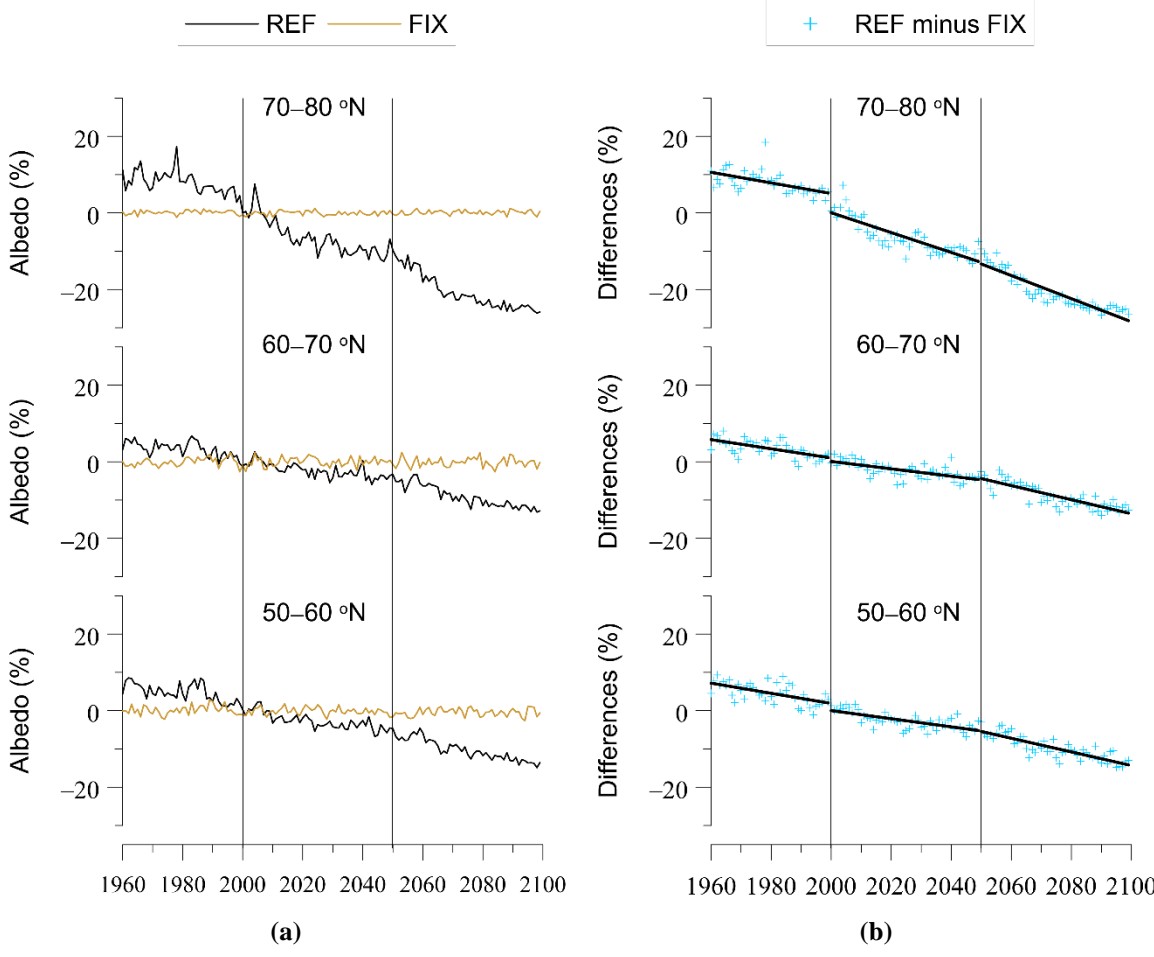

**Figure C1.** EMAC CCM projections of zonal mean surface albedo for 10-degree latitude zones (50-60º N, 60-70º N, 70-80º N), based on simulations with increasing and fixed GHGs mixing ratios. **(a)** REF is the simulation with increasing GHGs according to RCP-6.0. FIX is the simulation with fixed GHGs emissions at 1960 levels. **(b)** Difference between the two model simulations, as an indicator of the impact of increasing GHGs. The y-axis in the left figure (a) shows yearly averaged surface albedo data (in %) calculated from de-seasonalized monthly data. The monthly data were de-seasonalized relative to the long-term monthly mean (1990–2019) and were expressed in %. For the northern high latitudes, the annual average refers to the average of monthly anomalies from March to September.

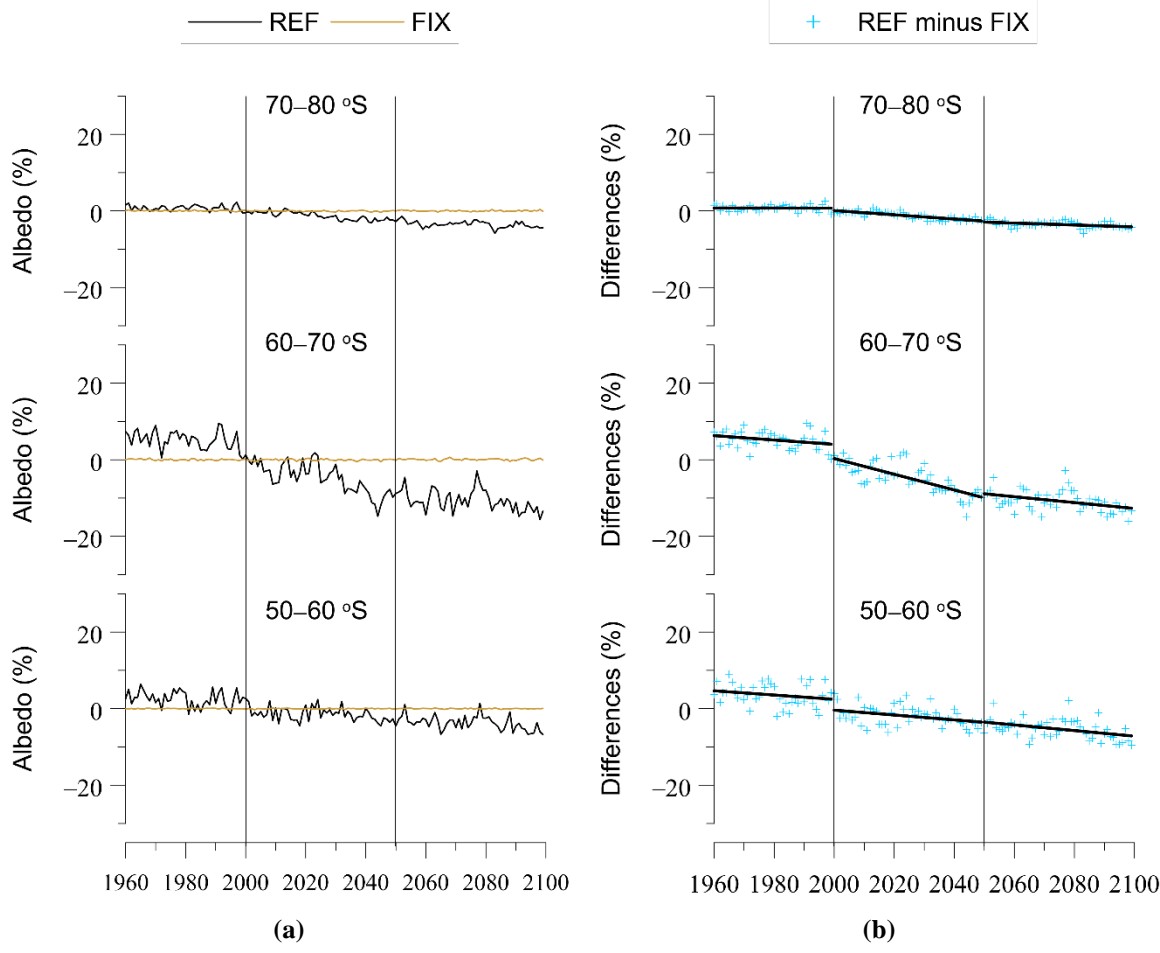

**Figure C2.** Same as Figure C1 but for 50-60º S, 60-70º S, and 70-80º S. The y-axis in the left figure (a) shows yearly averaged surface albedo data (in %) calculated from de-seasonalized monthly data. The monthly data were de-seasonalized relative to the long-term monthly mean (1990–2019) and were expressed in %. For the southern high latitudes, the annual average refers to the average of monthly anomalies from September to March.

**Data Availability:** The UV irradiance data are archived at the NDACC data repository, https://www-air.larc.nasa.gov/pub/NDACC/PUBLIC/stations/ (last access 4 September 2022). The SBUV (v8.7) satellite ozone data are available at https://acd-ext.gsfc.nasa.gov/Data_services/merged/previous_mods.html (last access 4 September 2022). The MODIS/Terra v6.1 satellite cloud fraction monthly mean data (MOD08_M3 v6.1) are available at https://giovanni.gsfc.nasa.gov/giovanni/#service=TmAvMp&starttime=&endtime=&data=MOD08_M3_6_1_Cloud_Fraction_Mean_Mean (last access 4 September 2022).

**Author Contribution:** K.E. and C.Z. conceptualized the study. A.B., G.B., D.K., S.S., B.L., C.B., F.A.,
S.S., H.D., and K.L. provided ground-based UV irradiance data. K.E., D.K., I.F. and K.T. analysed data.
M.D. and P.J. provided the EMAC model simulations. J.K. and K.D. processed the model simulations. The
manuscript was originally prepared by K.E. and was reviewed with comments and corrections from all co-
authors.
**Competing interests:** One co-author (MD) is coordinator, and one co-author (IP) is co-organizer of the
special issue "Atmospheric ozone and related species in the early 2020s: latest results and trends
(ACP/AMT inter-journal SI), 2021".

**Acknowledgments:** The research work was partially funded by the Hellenic Foundation for Research and
Innovation (H.F.R.I.) under the "First Call for H.F.R.I. Research Projects to support Faculty members and
Researchers and the procurement of high-cost research equipment grant" (Atmospheric parameters
affecting SPectral solar IRradiance and solar Energy (ASPIRE), Project Number: 300). We acknowledge
support by the project "PANhellenic infrastructure for Atmospheric Composition and climatE change"
(MIS 5021516) which is implemented under the Action "Reinforcement of the Research and Innovation
Infrastructure", funded by the Operational Programme "Competitiveness, Entrepreneurship and Innovation"
(NSRF 2014-2020), and co-financed by Greece and the European Union (European Regional Development
Fund). The research contributes to the National Network for Climate Change and its Impact – CLIMPACT.
We acknowledge the project Long-term Ozone Trends and Uncertainties in the Stratosphere (LOTUS) and
the Mariolopoulos-Kanaginis Foundation for the Environmental Sciences. The EMAC simulations have
been performed at the German Climate Computing Centre (DKRZ) through support from the
Bundesministerium für Bildung und Forschung (BMBF). DKRZ and its scientific steering committee are
gratefully acknowledged for providing the HPC and data archiving resources for this consortial project
ESCiMo (Earth System Chemistry integrated Modelling). Measurements of French spectroradiometers are
supported by CNES (French programme TOSCA); the Université de La Réunion and CNRS; the Région
Hauts-de-France and the Ministère de l'Enseignement Supérieur et de la Recherche (CPER Climibio); and
the European Fund for Regional Economic Development. Technicians at the three French and at the
Finnish (Sodankylä) sites are acknowledged for the maintenance and calibration of the instruments. Kaisa
Lakkala is supported by the CHAMPS project (grant no. 329225) of the Academy of Finland under the
CLIHE programme. We acknowledge the SBUV science team for providing the satellite ozone profiles.
NIWA UV spectrometer systems, serial numbers UV3 (Mauna Loa, HI) and UV5 (Boulder, CO) in the
USA are owned and operated by NOAA/ESRL/Global Monitoring Division, Boulder CO. They are
maintained, calibrated, and operated by NOAA. The final data from both of these instruments are quality
controlled and produced by NIWA-Lauder, New Zealand. We thank NIWA and NOAA for the use of their
data in this publication. Analyses and visualizations used in this study were produced with the Giovanni
online data system, developed and maintained by the NASA GES DISC. We also acknowledge the MODIS
mission scientists and associated NASA personnel for the production of the data used in this research
effort. We would like to thank the anonymous reviewer for summarizing the key findings which we
incorporated at the end of the conclusions.

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

**Table 1.** Ground-based stations with long-term UV measurements used for the evaluation of EMAC CCM
DNA active irradiance simulations. Stations are listed from northern to southern high latitudes and are
grouped as follows: 3 stations at latitudes greater than 55° N, 13 stations between 50° N – 50° S and 4
stations at latitudes greater than 55° S.

| Station name | Latitude | Longitude | Period |
|---|---|---|---|
| 1. Summit, Greenland* | 72.58 | -38.45 | 08/2004-08/2017 |
| 2. Barrow, AK, United States* | 71.32 | -156.68 | 02/1991-07/2016 |
| 3. Sodankylä, Finland | 67.37 | 26.63 | 01/1990-12/2021 |
| 4. Villeneuve d'Ascq, France* | 50.61 | 3.14 | 01/2000-12/2019 |
| 5. Groß-Enzersdorf, Austria* | 48.20 | 16.56 | 05/1998-11/2019 |
| 6. Zugspitze, Germany* | 47.42 | 10.98 | 08/1995-06/2007 |
| 7. Hoher Sonnblick, Austria* | 47.05 | 12.95 | 01/1997-06/2020 |
| 8. Aosta, Italy | 45.74 | 7.36 | 08/2006-09/2020 |
| 9. Observatoire de Haute Provence, France* | 43.94 | 5.70 | 01/2009-11/2018 |
| 10. Thessaloniki, Greece | 40.63 | 22.95 | 08/1993-12/2019 |
| 11. Boulder, CO, United States* | 39.99 | -105.26 | 01/2004-12/2019 |
| 12. Athens, Greece | 37.99 | 23.78 | 07/2004-12/2020 |
| 13. Mauna Loa, HI, United States* | 19.53 | -155.58 | 07/1995-12/2019 |
| 14. Reunion Island, St. Denis, France* | -20.90 | 55.50 | 03/2009-12/2019 |
| 15. Alice Springs, Australia* | -23.80 | 133.87 | 01/2005-12/2019 |
| 16. Lauder, New Zealand* | -45.04 | 169.68 | 01/1991-12/2019 |
| 17. Ushuaia, Argentina* | -54.82 | -68.32 | 01/1990-11/2008 |
| 18. Palmer, Antarctica* | -64.77 | -64.05 | 03/1990-05/2021 |
| 19. Arrival Heights, Antarctica* | -77.83 | 166.67 | 01/1990-04/2021 |
| 20. South Pole, Antarctica* | -90 | 0 | 11/1990-03/2021 |

*NDACC sites


**Table 2. (a)** Correlation results between model simulations (HIS-SD) and ground-based DNA active
irradiance data for the northern high latitude stations (>55º N), the southern high latitude stations (>55º S),
and the stations between 50º N – 50º S. **(b)** Same as (a) but for the HIS-SD simulation and satellite SBUV
(v8.7) total ozone data. **(c)** Same as (a) but for the HIS-SD simulation and satellite MODIS/Terra cloud
fraction data. Error, t-value, and p-value refer to slope; t-value should be higher than 2.576 for 99%
statistical significance.

| (a) DNA active irradiance | | | | | | | |
|---|---|---|---|---|---|---|---|
| | R | Slope | Error | t-value | p-value | N | RMSE |
| >55º N | +0.504 | 0.807 | 0.125 | 6.447 | <0.0001 | 124 | 11.527 |
| >55º S | +0.746 | 0.879 | 0.070 | 12.629 | <0.0001 | 129 | 14.766 |
| 50º N – 50º S | +0.499 | 0.387 | 0.045 | 8.564 | <0.0001 | 223 | 4.215 |
| | | | | | | | |
| (b) Total ozone | | | | | | | |
| | R | Slope | Error | t-value | p-value | N | RMSE |
| >55º N | +0.899 | 0.858 | 0.037 | 23.264 | <0.0001 | 131 | 1.308 |
| >55º S | +0.892 | 0.888 | 0.040 | 22.211 | <0.0001 | 129 | 3.414 |
| 50º N – 50º S | +0.894 | 0.817 | 0.028 | 29.672 | <0.0001 | 223 | 0.872 |
| | | | | | | | |
| (c) Cloud cover | | | | | | | |
| | R | Slope | Error | t-value | p-value | N | RMSE |
| >55º N | +0.453 | 0.362 | 0.063 | 5.775 | <0.0001 | 131 | 4.483 |
| >55º S | +0.485 | 0.806 | 0.129 | 6.230 | <0.0001 | 128 | 5.003 |
| 50º N – 50º S | +0.703 | 0.721 | 0.049 | 14.674 | <0.0001 | 222 | 5.162 |





**Table 3.** Trends (% per decade) in total ozone, DNA active irradiance, and cloudiness from the two model
simulations and the differences between them, i.e., free-running simulation with increasing GHGs (REF)
minus the simulation with fixed GHGs at 1960 levels (FIX), averaged at 3 stations in the northern high
latitudes (>55° N), 4 stations in the southern high latitudes (>55° S), and 13 stations between 50° N – 50° S.
The trends are estimated from the annual mean anomalies shown in Figures 4, 5, 6.

| **>55° N (3 stations)** | | | | | | | | | |
|---|---|---|---|---|---|---|---|---|---|
| **Trends (% per decade)** | **REF** | | | **FIX** | | | **Difference** | | |
| | **1960-1999** | **2000-2049** | **2050-2099** | **1960-1999** | **2000-2049** | **2050-2099** | **1960-1999** | **2000-2049** | **2050-2099** |
| Ozone | −0.55 ± 0.21 | 0.50 ± 0.16 | 0.41 ± 0.17 | −1.35 ± 0.26 | 0.36 ± 0.15 | 0.05 ± 0.15 | 0.79 ± 0.24 | 0.14 ± 0.16 | 0.36 ± 0.17 |
| DNA active irradiance | 1.03 ± 0.77 | −1.75 ± 0.56 | −2.47 ± 0.60 | 2.61 ± 0.78 | −1.15 ± 0.46 | −0.83 ± 0.57 | −1.57 ± 0.96 | −0.59 ± 0.68 | −1.64 ± 0.76 |
| Clouds | −0.21 ± 0.30 | −0.10 ± 0.23 | 0.32 ± 0.20 | −0.20 ± 0.43 | 0.02 ± 0.25 | 0.19 ± 0.27 | −0.29 ± 0.45 | −0.39 ± 0.25 | 0.29 ± 0.27 |
| **>55° S (4 stations)** | | | | | | | | | |
| **Trends (% per decade)** | **REF** | | | **FIX** | | | **Difference** | | |
| | **1960-1999** | **2000-2049** | **2050-2099** | **1960-1999** | **2000-2049** | **2050-2099** | **1960-1999** | **2000-2049** | **2050-2099** |
| Ozone | −3.78 ± 0.57 | 2.63 ± 0.31 | 1.42 ± 0.29 | −4.65 ± 0.48 | 2.28 ± 0.41 | 0.58 ± 0.31 | 0.87 ± 0.56 | 0.35 ± 0.46 | 0.84 ± 0.42 |
| DNA active irradiance | 5.70 ± 0.97 | −4.92 ± 0.55 | −1.68 ± 0.43 | 7.61 ± 0.92 | −4.81 ± 0.75 | −0.72 ± 0.38 | −1.91 ± 0.95 | −0.10 ± 0.82 | −0.97 ± 0.58 |
| Clouds | −0.36 ± 0.39 | −0.71 ± 0.28 | −0.28 ± 0.26 | 0.18 ± 0.31 | −0.05 ± 0.22 | −0.01 ± 0.26 | −0.54 ± 0.47 | −0.53 ± 0.35 | −0.21 ± 0.35 |
| **50° N – 50° S (13 stations)** | | | | | | | | | |
| **Trends (% per decade)** | **REF** | | | **FIX** | | | **Difference** | | |
| | **1960-1999** | **2000-2049** | **2050-2099** | **1960-1999** | **2000-2049** | **2050-2099** | **1960-1999** | **2000-2049** | **2050-2099** |
| Ozone | −0.61 ± 0.14 | 0.42 ± 0.11 | 0.02 ± 0.12 | −1.27 ± 0.16 | 0.36 ± 0.10 | 0.12 ± 0.09 | 0.66 ± 0.10 | 0.06 ± 0.08 | −0.10 ± 0.09 |
| DNA active irradiance | 1.55 ± 0.44 | −0.53 ± 0.35 | 0.86 ± 0.35 | 1.75 ± 0.48 | −0.54 ± 0.32 | 0.05 ± 0.28 | −0.20 ± 0.58 | 0.01 ± 0.44 | 0.81 ± 0.39 |
| Clouds | −1.15 ± 0.29 | −0.30 ± 0.27 | −0.60 ± 0.25 | 0.16 ± 0.30 | −0.25 ± 0.23 | −0.16 ± 0.19 | −1.21 ± 0.42 | −0.12 ± 0.30 | −0.50 ± 0.27 |




**Table 4.** Same as Table 3 but for the winter months, January (J), February (F), and March (M) for the
northern high latitude stations. Due to the polar night, UV results for January and February are not shown
due to large standard errors.

<table>
<tr><td colspan="10" align="center">>55° N (3 stations)</td></tr>
<tr><td rowspan="2">Trends (%<br>per decade)</td><td colspan="3" align="center">REF</td><td colspan="3" align="center">FIX</td><td colspan="3" align="center">Difference</td></tr>
<tr><td>1960-1999</td><td>2000-2049</td><td>2050-2099</td><td>1960-1999</td><td>2000-2049</td><td>2050-2099</td><td>1960-1999</td><td>2000-2049</td><td>2050-2099</td></tr>
<tr><td>Ozone (J)</td><td>−1.66 ± 0.74</td><td>1.81 ± 0.53</td><td>0.49 ± 0.53</td><td>−1.32 ± 0.72</td><td>0.28 ± 0.50</td><td>0.28 ± 0.54</td><td>−0.34 ± 1.09</td><td>1.53 ± 0.64</td><td>0.21 ± 0.73</td></tr>
<tr><td>Ozone (F)</td><td>−2.27 ± 0.74</td><td>2.00 ± 0.59</td><td>0.88 ± 0.58</td><td>−1.65 ± 0.83</td><td>0.21 ± 0.54</td><td>0.31 ± 0.44</td><td>−0.62 ± 0.96</td><td>1.79 ± 0.78</td><td>0.58 ± 0.71</td></tr>
<tr><td>Ozone (M)</td><td>−2.98 ± 0.53</td><td>0.75 ± 0.50</td><td>0.66 ± 0.35</td><td>−3.04 ± 0.71</td><td>0.58 ± 0.39</td><td>−0.54 ± 0.41</td><td>0.06 ± 0.83</td><td>0.17 ± 0.58</td><td>1.20 ± 0.51</td></tr>
<tr><td>DNA active<br>irradiance (J)</td><td>Polar night</td><td>Polar night</td><td>Polar night</td><td>Polar night</td><td>Polar night</td><td>Polar night</td><td>Polar night</td><td>Polar night</td><td>Polar night</td></tr>
<tr><td>DNA active<br>irradiance (F)</td><td>Polar night</td><td>Polar night</td><td>Polar night</td><td>Polar night</td><td>Polar night</td><td>Polar night</td><td>Polar night</td><td>Polar night</td><td>Polar night</td></tr>
<tr><td>DNA active<br>irradiance (M)</td><td>3.03 ± 1.97</td><td>−0.63 ± 1.52</td><td>−2.83 ± 1.22</td><td>7.20 ± 1.77</td><td>−0.09 ± 1.44</td><td>−0.98 ± 1.27</td><td>−3.28 ± 2.55</td><td>−0.06 ± 2.18</td><td>−1.90 ± 1.51</td></tr>
<tr><td>Clouds (J)</td><td>−0.67 ± 0.61</td><td>1.29 ± 0.63</td><td>0.94 ± 0.48</td><td>1.11 ± 0.77</td><td>0.03 ± 0.53</td><td>0.52 ± 0.49</td><td>−1.77 ± 0.92</td><td>1.25 ± 0.88</td><td>0.39 ± 0.68</td></tr>
<tr><td>Clouds (F)</td><td>−0.07 ± 0.86</td><td>0.60 ± 0.68</td><td>1.30 ± 0.57</td><td>0.55 ± 0.81</td><td>−0.14 ± 0.66</td><td>0.06 ± 0.58</td><td>−0.62 ± 1.11</td><td>0.74 ± 0.93</td><td>1.24 ± 0.82</td></tr>
<tr><td>Clouds (M)</td><td>0.35 ± 0.70</td><td>1.15 ± 0.60</td><td>1.33 ± 0.61</td><td>0.23 ± 1.02</td><td>0.21 ± 0.69</td><td>−0.17 ± 0.64</td><td>0.12 ± 1.12</td><td>0.94 ± 0.88</td><td>1.49 ± 0.79</td></tr>
</table>




**Table 5.** Statistical test results for the difference between two trends in DNA active irradiance (trend of
1960-2049 minus trend of 2050-2099), for the northern high latitude stations (>55º N), the southern high
latitude stations (>55º S), and the stations between 50º N – 50º S.

| Latitudes | >55º N (3 stations) | | >55º S (4 stations) | | 50º N – 50º S (13 stations) | |
|---|---|---|---|---|---|---|
| | 1960–2049 | 2050–2099 | 1960–2049 | 2050–2099 | 1960–2049 | 2050–2099 |
| $N$ | 553 | 315 | 630 | 350 | 1080 | 600 |
| slope, $b/year$ (Eq. 4) | –0.173 | –0.186 | –0.116 | –0.096 | –0.033 | 0.081 |
| $s_b$ (Eqs. 5 and 6) | 0.027 | 0.062 | 0.026 | 0.048 | 0.015 | 0.037 |
| $s_{(b_1-b_2)}$ (Eq. 3) | 0.068 | | 0.054 | | 0.040 | |
| $t-value$ (Eq. 3) | 0.185 | | –0.376 | | –2.844 | |
| degrees of freedom | 864 | | 976 | | 1676 | |
| significance level | 0.05 | | 0.05 | | 0.05 | |
| $p-value$ | 0.853 | | 0.707 | | 0.005 | |
| $t-critical$ | 1.96 | | 1.96 | | 1.96 | |
| Significantly different trends | No | | No | | Yes | |




**Table 6. (a)** Coefficients of multiple regression analysis according to Eq. (7), applied to the differences
between the two model simulations, REF and FIX, for the period 2050–2099, for the northern high latitude
stations (>55$^o$ N), the southern high latitude stations (>55$^o$ S), and the stations between 50$^o$ N – 50$^o$ S. **(b)**
Trends (% per decade) for the period 2050-2099 in the DNA active irradiance, the ozone-related DNA
active irradiance component and the cloud-related DNA active irradiance component.

| **(a) MLR coefficients (2050–2099)** | | | |
|---|---|---|---|
| | **>55$^o$ N (3 stations)** | **>55$^o$ S (4 stations)** | **50$^o$ N – 50$^o$ S (13 stations)** |
| $a \pm error$ | −4.473 ± 0.976 | −1.994 ± 0.670 | −0.557 ± 0.336 |
| $\beta_{O_3} \pm error$ | −2.017 ± 0.220 | −0.667 ± 0.071 | −2.831 ± 0.128 |
| $\beta_{cloud} \pm error$ | −0.749 ± 0.090 | −0.367 ± 0.065 | −0.642 ± 0.035 |
| | | | |
| **(b) Trends (% per decade) (2050–2099)** | | | |
| | **>55$^o$ N (3 stations)** | **>55$^o$ S (4 stations)** | **50$^o$ N – 50$^o$ S (13 stations)** |
| DNA active irradiance | −1.86 ± 0.61% | −0.96 ± 0.48% | 0.81 ± 0.37% |
| Ozone-related DNA active irradiance component | −0.72 ± 0.25% | −0.57 ± 0.21% | 0.27 ± 0.20% |
| Cloud-related DNA active irradiance component | −0.21 ± 0.24% | 0.07 ± 0.13% | 0.33 ± 0.17% |




**Table 7.** Trends and their standard errors (% per decade) in the differences between the two model
simulations, REF and FIX, for the DNA active irradiance, total ozone, cloud cover and surface albedo at
Barrow (Alaska) and Palmer (Antarctica) for the periods 1960–1999, 2000–2049 and 2050–2099.

| Trends (% per decade) | Barrow, Alaska | | | Palmer, Antarctica | | |
|---|---|---|---|---|---|---|
| | **1960-1999** | **2000-2049** | **2050-2099** | **1960-1999** | **2000-2049** | **2050-2099** |
| DNA active irradiance | −2.88 ± 1.67 | −2.18 ± 1.17 | −2.14 ± 1.12 | 0.75 ± 1.47 | −1.79 ± 1.08 | −0.33 ± 0.90 |
| Ozone | 0.39 ± 0.24 | 0.06 ± 0.17 | 0.44 ± 0.19 | −0.02 ± 0.54 | 0.23 ± 0.37 | 0.54 ± 0.40 |
| Clouds | −0.96 ± 0.78 | 0.42 ± 0.54 | 0.60 ± 0.52 | −1.60 ± 0.65 | 0.41 ± 0.49 | −0.46 ± 0.48 |
| Surface albedo | 0.88 ± 1.33 | −6.42 ± 0.80 | −2.73 ± 0.89 | 0.08 ± 0.82 | −1.52 ± 0.51 | −1.60 ± 0.53 |





DNA active irradiance comparisons
── EMAC CCM (HIS-SD)
── Ground-based measurements



**Figure 1.** Comparison of model simulations of DNA active irradiance with averages of ground-based measurements at 3 UV stations in the northern high latitudes (>55º N) (upper panel), 13 UV stations from 50º N to 50º S (middle panel) and 4 UV stations in the southern high latitudes (>55º S) (lower panel). The y-axis shows monthly de-seasonalized DNA active irradiance data (in %). The monthly data at each station were de-seasonalized by subtracting the long-term monthly mean (2000–2018) pertaining to the same calendar month and were expressed in %. Then, the average over each geographical zone was estimated by averaging the de-seasonalized data of the stations belonging to each geographical zone. Shown are data from March to September for the northern high latitudes and from September to March for the southern high latitudes.


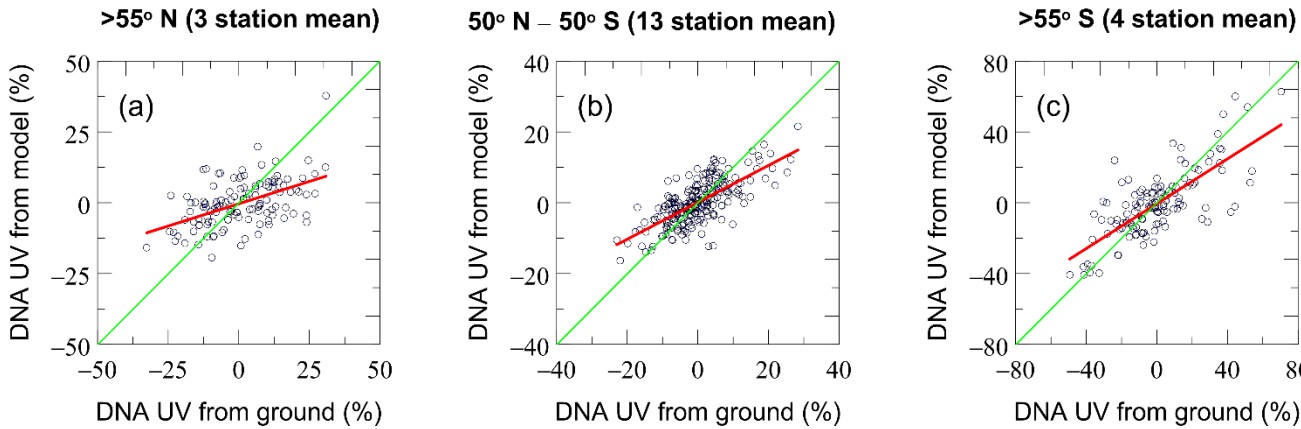



**Figure 2.** Scatter plots of DNA active irradiance from simulated and ground-based data shown in Figure 1
for **(a)** 3 UV stations in the northern high latitudes (>55º N), **(b)** 13 UV stations from 50º N to 50º S and **(c)**
4 UV stations in the southern high latitudes (>55º S).




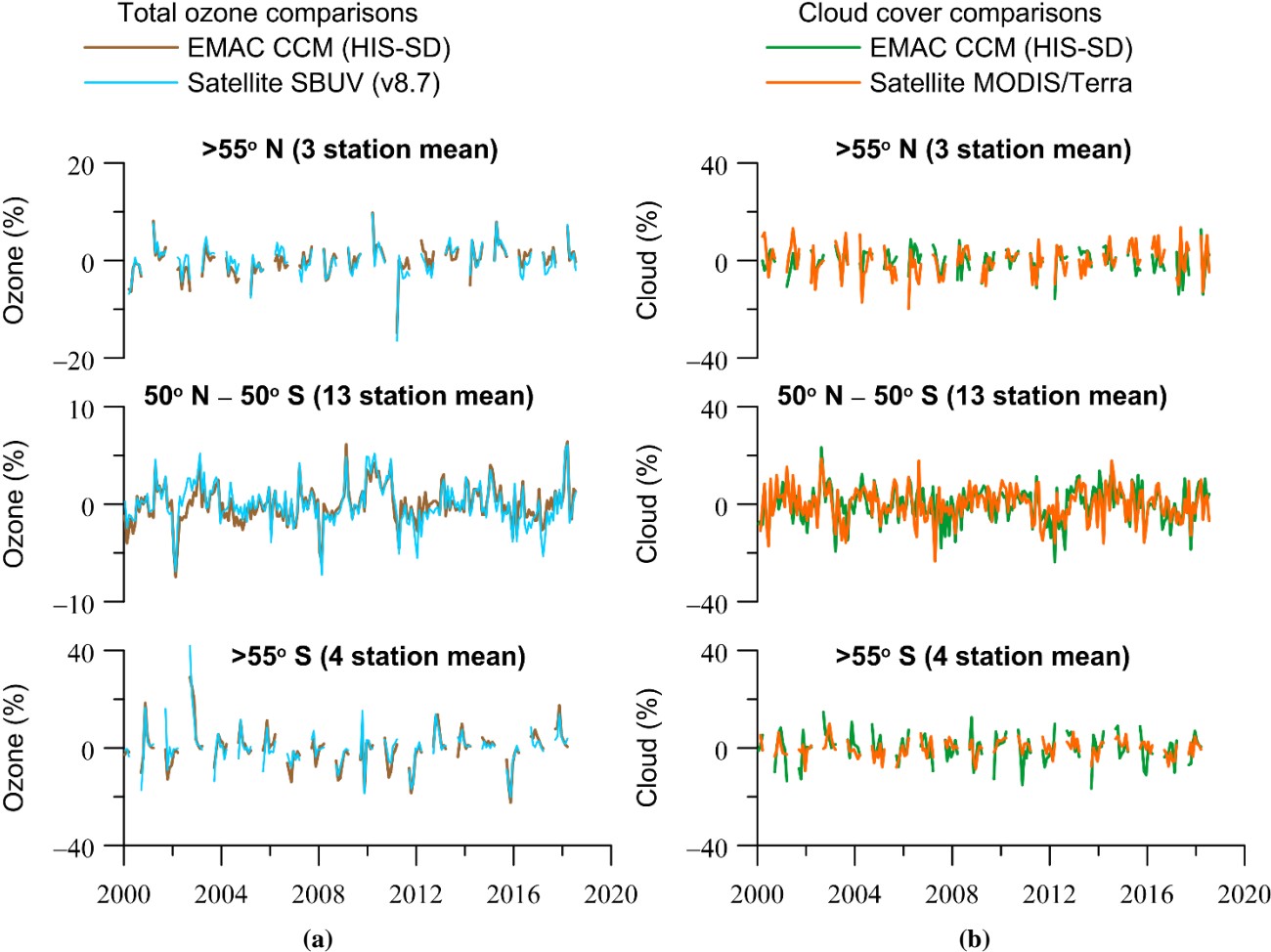

(a)

(b)


**Figure 3. (a)** Same as in Figure 1 but for ozone column. **(b)** For cloud cover. The y-axes show monthly de-
seasonalized anomalies (in %) relative to the long-term monthly mean (2000–2018). Shown are monthly
anomalies from March to September for the northern high latitudes, and from September to March for the
southern high latitudes. For 50° N–50° S, we present all months.

1262

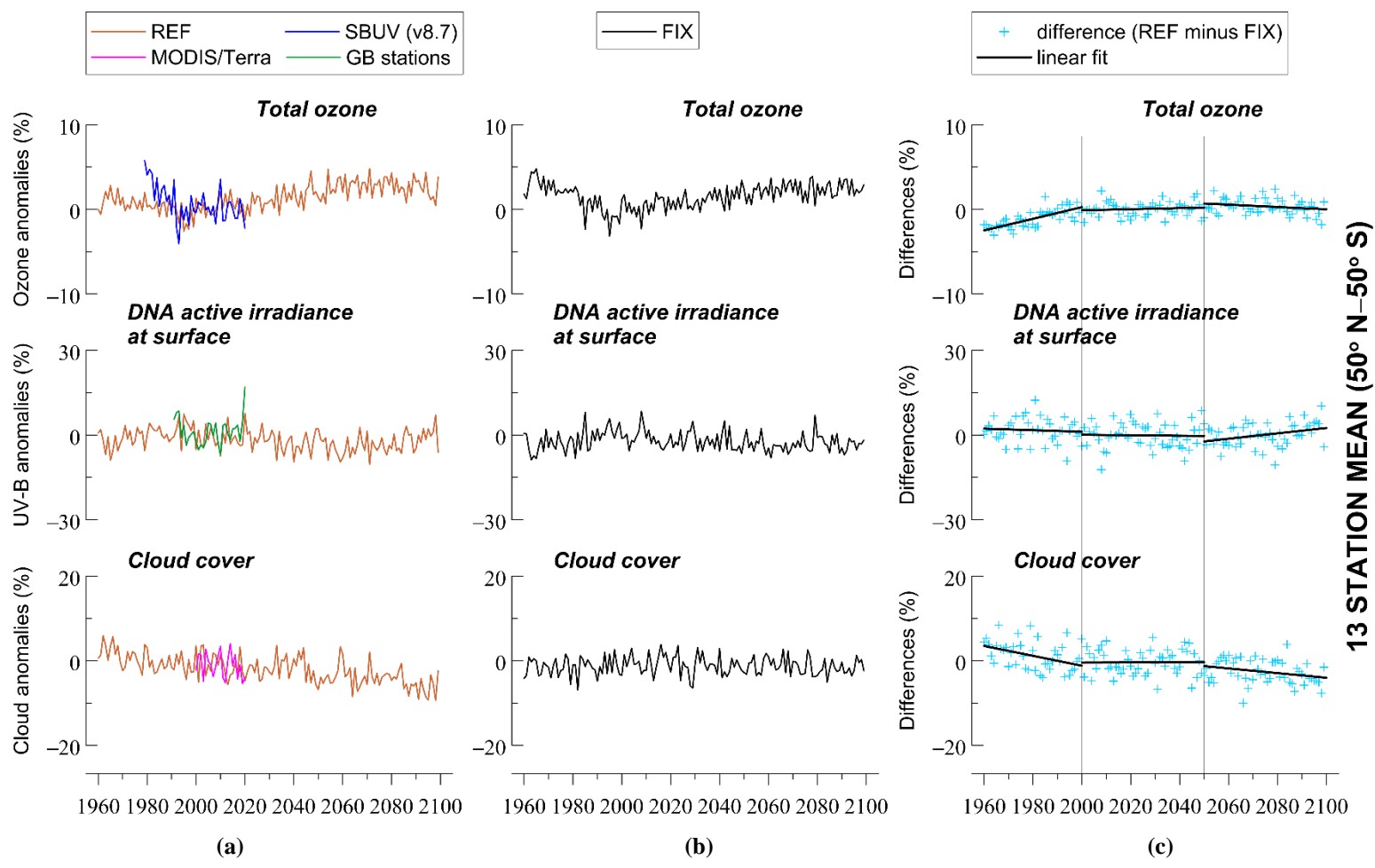

1263

**Figure 4.** Changes in total ozone, DNA active irradiance and cloud cover averaged at 13 UV stations from 50º N to 50º S, based on simulations with increasing and fixed GHGs mixing ratios. **(a)** REF is the simulation with increasing GHGs according to RCP-6.0. **(b)** FIX is the simulation with fixed GHGs emissions at 1960 levels. **(c)** Difference between the two model simulations, indicating the impact of increasing GHGs. The y-axes in (a) and (b) show yearly averaged data (in %) calculated from de-seasonalized monthly data. The monthly data were de-seasonalized relative to the long-term monthly mean (1990–2019) and were expressed in %. For stations between 50º N–50º S we used all months to calculate the annual average.


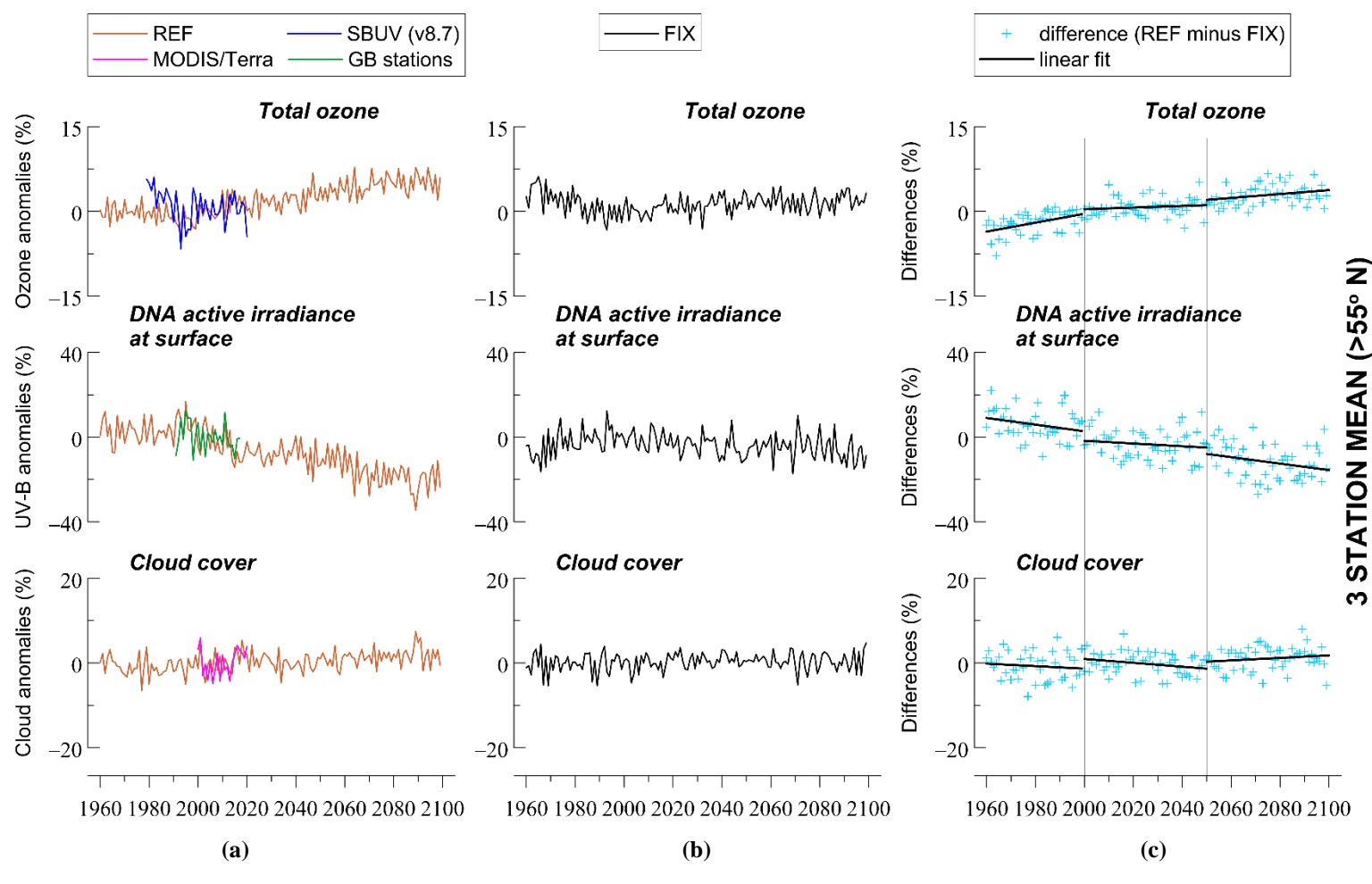


**Figure 5.** Same as Figure 4 but for 3 UV stations in the northern high latitudes (>55° N). The y-axes in (a)
and (b) show averages of monthly de-seasonalized anomalies from March to September.

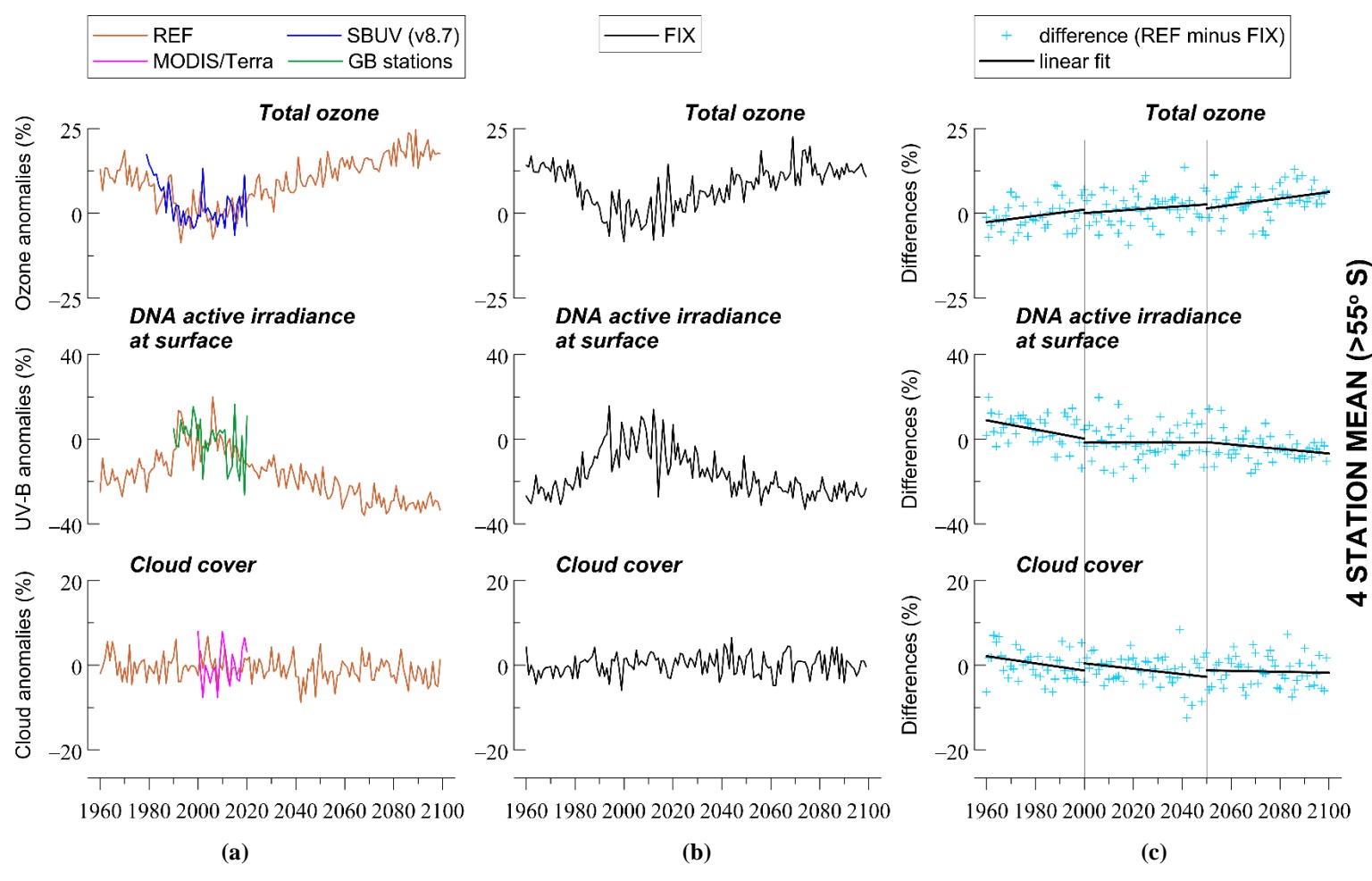


**Figure 6.** Same as Figure 4 but for 4 UV stations in the southern high latitudes (>55° S). The y-axes in (a)
and (b) show averages of monthly de-seasonalized anomalies from September to March.


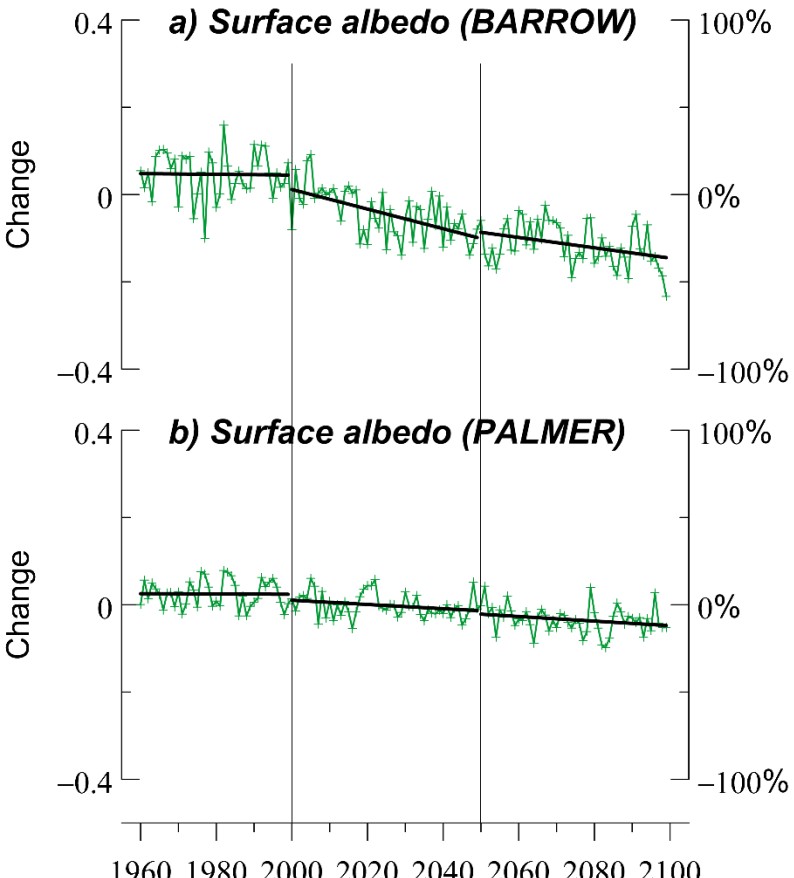



**Figure 7. (a)** Changes in surface albedo at Barrow, Alaska, and **(b)** at Palmer, Antarctica, derived from the
differences between the two model simulations: the one with increasing GHGs (REF) and the one with
fixed GHGs (FIX). Results refer to the summer season. Data were de-seasonalized with respect the period
1990–2019 and then were averaged from March to September at Barrow, and from September to March at
Palmer. The left y-axis shows the differences in surface albedo values and the right y-axis shows the
respective differences in % of the mean.
