# Peer review of "Ozone, DNA active UV radiation and cloud changes for the"

_EGUsphere, 2022_

## Referee Comment (RC1)

Review of the manuscript "Ozone and DNA active UV radiation changes for the near global mean and at high latitudes due to enhanced greenhouse gas concentrations", by Eleftheratos et al.

The manuscript presents results from the global chemistry climate model ECAM on changes regarding DNA weighted UV radiation with regard to changes in total ozone, cloud cover and surface albedo. UV data derived from the model are validated with respect to past UV data from 13 ground-based stations providing solar UV measurements while the simulated ozone is compared to SBUV ozone.

The results are discussed as three groups, defined by North and South high latitudes and middle latitudes, and model results are retrieved at the locations of the 13 ground-based stations.

The key findings presented in the manuscript are that model and measurements agree fairly well, giving support to the simulations of the future scenarios. Cloud cover is generally decreasing, leading to increased solar radiation, apart from the high latitudes, where no significant changes are observed. UV trends are a combination of ozone changes (mostly ozone recovery), and cloud cover changes, while at high latitudes, decreased surface albedo in the second half of the next century have a significant influence on the surface UV radiation.

The manuscript is well written, the references are extensive and cover the current status of the field as far as I can judge. The results are interesting and therefore the manuscript is in principle worth to be published.

However I have serious concerns with the novelty of the research and its added value with respect to already published papers, foremost the one published in 2020 by the same main author, Eleftheratos, K., Kapsomenakis, J., Zerefos, C. S., Bais, A. F., Fountoulakis, I., Dameris, M., Jöckel, P., 821 Haslerud, A. S., Godin-Beekmann, S., Steinbrecht, W., Petropavlovskikh, I., Brogniez, C., Leblanc, T., Liley, 822 J. B., Querel R., and Swart, D. P. J.: Possible Effects of Greenhouse Gases to Ozone Profiles and DNA Active 823 UV-B Irradiance at Ground Level, Atmosphere, 11, 228, doi:10.3390/atmos11030228, 2020.

The authors discusses this manuscript at length, so they are aware that there is a need for distinction. However it seems that the main difference in this manuscript with respect to the previous work are the addition of a few ground-based stations at which the model results are analysed (13 instead of 5, of which 4 are identical). The conclusions of the manuscript are very similar to the previous manuscript, with some differences by distinguishing three latitudinal bands.

Since the model used in the analysis is a global model, the restriction to 13 specific sites must be for good reason. The only reason I can see is that this allows comparison between the ground based UV stations with the model at these locations, in order to validate the model. I am not convinced by this argument for the following reasons:

- The model has a resolution of 3°x3°, which is a huge area for which the point measurement has to be representative for. I doubt that this is the case for several stations where the surrounding area is inhomogeneous, such as mountain tops (MaunaLoa, Sonnblick, Zugspitze), or in valleys (Aosta), or by being in a town with very heterogeneous surroundings (sea, mountains, tropospheric ozone and aerosols such as Athens).
- The correlation for DNA weighted UV irradiance is actually not very good, and the figures in the supplementary material show quite different behaviour. A general comment is that the correlation is not the only measure for the agreement between two datasets, but also the slope between two datasets (from a scatter plot), are significantly different from one,

showing that the model results disagree quite significantly from the measurements, see Table S1 with a summary of the statistics. For me these comparisons do not support any validation of the model.

- The comparisons were performed for past to present data, using a model with prescribed dynamics. However the results of the manuscript are obtained using a free-running model, which as the authors write themselves, has serious shortcomings. The Appendix A discusses this fact, which is appreciated.
- ozone and Cloud cover are obtained from satellite measurements, which also give a global product.
- The authors themselves mention that some stations might not be very representative, being close to the shore (Barrow and Palmer, line 493).

Therefore the benefit of restricting the analysis of the model results to only 13 point locations does not compensate for the results obtained if the model results were analysed as a whole, for example in latitudinal bands, or by selecting specific regions where future changes are expected to be very different (Europe versus Asia, Sahara, …).

Some specific comments:

- The simulations of the future climate did not take into account possible solar variabilities (grand minimum, as discussed in Anet, J., Rozanov, S. Muthers, et al., Impact of a potential 21st century "grand solar minimum" on surface temperatures and stratospheric ozone, Geophys. Res. Lett., 40, 4420–4425, doi:10.1002/grl.50806, 2013, and Arsenovic, P., Rozanov, J. Anet, et al., Implications of potential future grand solar minimum for ozone layer and climate, Atmos. Chem. Phys., 18, 3469–3483, doi: 10.5194/acp-18-3469-2018, 2018.
- Future changes in aerosol loading are expected to be significant in some areas of the globe, having a strong impact on the UV radiation reaching the surface.
- The significance of the results are mainly described by correlation coefficient and p values. The supporting figures, for example Figure 1, however show that the variability between the model and ground based stations is very large. Even though scatter plots are also not the method of choice, they would give a better indication how two datasets would scatter, and the slope and associated fitting uncertainties would give an indication on how well the two datasets agree. I would have preferred the authors to have used other metrics as well, such as uncertainties (at the 95% confidence level) derived from the statistical models.
- The statistical approach of using a MLR technique is interesting, but why did not the authors include in equation 7 also the surface albedo, instead of treating it separately in the following section?
- The datasets would have been ideally suited to be analysed using the very powerful Dynamical Linear Modelling (DLR), for example, Alsing, (2019) dlmmc: Dynamical linear model regression for atmospheric time-series analysis. Journal of Open Source Software, 4(37), 1157, https://doi.org/10.21105/joss.01157, and Laine, M., Latva-Pukkila, N., and Kyrölä, E.: Analysing time-varying trends in stratospheric ozone time series using the state space approach, Atmos. Chem. Phys., 14, 9707–9725, https://doi.org/10.5194/acp-14-9707-2014, 2014.
- Figure 1: 50N-50S, there is a striking difference between model and measurements around the year 2012, of about 15%, which seem not be seen in either ozone or cloud cover. Did the authors investigate this feature?
- In the figures from the supplement, the model variabilities of the DNA weighted irradiance are much larger than the corresponding measurements for : Barrow, Villeneuve d'Ascq,

Aosta, Lauder, Ushuaia, while they are in better agreement for Summit, Thessaloniki, Boulder, MaunaLoa, or Alice Springs. For Athens, between approx. 2012 and 2015 the measurements are significantly higher than the model results, why is that so?

– The cloud cover from Modis/Terra and the model show no correlation for most stations, apart for example for Aosta, which is slightly better. Can the authors provide some comments why some stations show better agreement than others?

---

## Author Comment (AC1)

**Answers to the Reviewers' comments**

We would like to thank the reviewers for carefully reading the manuscript and for their constructive comments, which helped us to improve our work. The comments are written with blue colour in Italic font. Our answers are given with normal font.

**Reply to Reviewer #1**

*Comments*

*The manuscript presents results from the global chemistry climate model ECAM on changes regarding DNA weighted UV radiation with regard to changes in total ozone, cloud cover and surface albedo. UV data derived from the model are validated with respect to past UV data from 13 ground-based stations providing solar UV measurements while the simulated ozone is compared to SBUV ozone.*

*The results are discussed as three groups, defined by North and South high latitudes and middle latitudes, and model results are retrieved at the locations of the 13 ground-based stations.*

*The key findings presented in the manuscript are that model and measurements agree fairly well, giving support to the simulations of the future scenarios. Cloud cover is generally decreasing, leading to increased solar radiation, apart from the high latitudes, where no significant changes are observed. UV trends are a combination of ozone changes (mostly ozone recovery), and cloud cover changes, while at high latitudes, decreased surface albedo in the second half of the next century have a significant influence on the surface UV radiation.*

*The manuscript is well written, the references are extensive and cover the current status of the field as far as I can judge. The results are interesting and therefore the manuscript is in principle worth to be published.*

**Answer:**

We thank the reviewer for the comments. We find very useful the reviewer's view of the key findings of our work, and we have incorporated them in the conclusions together with an appropriate acknowledgement in the Acknowledgements. We have answered all comments and made the appropriate changes to the manuscript. Important updates are that we added UV irradiance data from Sodankylä station in the plots for the

northern high latitudes (3 stations now instead of 2), and we prepared new figures using zonally averaged data for the three latitudinal bands.

In brief, the figures in the revised manuscript are as follows:

- Figure 1. DNA active irradiance (station based). Same as before, but now for 3 stations in the northern high latitudes instead of 2. Notation of simulations changed.
- Figure 2. Scatter plots of DNA active irradiance data (station-based) from Figure 1. New figure.
- Figure 3. Ozone and clouds (station based). Same as before, but now for 3 stations in the northern high latitudes instead of 2. Notation of simulations changed.
- Figure 4. Time series for the near global mean (station based). Same as before. Notation of simulations changed.
- Figure 5. Time series for the northern high latitudes (station based). Same as before, but now for 3 stations in the northern high latitudes instead of 2. Notation of simulations changed.
- Figure 6. Time series for the southern high latitudes (station based). Same as before. Notation of simulations changed.
- Figure 7. Time series of surface albedo for Barrow and Palmer. Same as before. Notation of simulations changed.

The new figures showing all latitude averaged results are presented in the updated Supplement as follows:

- Figure S1. Ozone and clouds from the global products (zonal based). New figure.
- Figure S2. Time series for the near global mean (zonal based). New figure.
- Figure S3. Time series for the northern high latitudes (zonal based). New figure.
- Figure S4. Time series for the southern high latitudes (zonal based). New figure.

*However I have serious concerns with the novelty of the research and its added value with respect to already published papers, foremost the one published in 2020 by the same main author, Eleftheratos, K., Kapsomenakis, J., Zerefos, C. S., Bais, A. F., Fountoulakis, I., Dameris, M., Jöckel, P., 821 Haslerud, A. S., Godin-Beekmann, S., Steinbrecht, W., Petropavlovskikh, I., Brogniez, C., Leblanc, T., Liley, 822 J. B., Querel R., and Swart, D. P. J.: Possible Effects of Greenhouse Gases to Ozone Profiles and DNA*

*Active 823 UV-B Irradiance at Ground Level, Atmosphere, 11, 228, doi:10.3390/atmos11030228, 2020.*

*The authors discuss this manuscript at length, so they are aware that there is a need for distinction. However it seems that the main difference in this manuscript with respect to the previous work are the addition of a few ground-based stations at which the model results are analysed (13 instead of 5, of which 4 are identical). The conclusions of the manuscript are very similar to the previous manuscript, with some differences by distinguishing three latitudinal bands.*

**Answer:**

The reviewer raised serious concerns about the novelty of the research and its added value with respect to the previous study by Eleftheratos et al., pointing out that there is need for distinction. We have added the following text in the Introduction, in order to distinct the new study from the previous one:

"It is important to clarify the novelty of this research and its added value with respect to the previous study, and to point out the main differences and similarities. The objective of this research is to study how total ozone, DNA active irradiance and cloud cover might change in the future at high latitudes due to the increasing GHGs in comparison to the near global mean (50º N–50º S). Also, to estimate the part of the DNA active irradiance change that can be explained by ozone and cloud changes in the future using multiple linear regression (MLR) statistical analysis. The previous study by Eleftheratos et al. (2020) did not look at high latitudes and did not apply MLR analysis to quantify the related contributions to the DNA weighted UV irradiance. The previous work analysed 5 stations between 50º N and 50º S. The new study is enriched with more stations at the near global scale (13 instead of 5 stations, of which 4 are identical) and, in addition, it includes analysis of averages in latitudinal bands, which was not done in the previous study, thus providing more complete results".

We hope that these clarifications will satisfy the reviewer's concerns about the novelty of the study.

The reviewer also commented that *"the conclusions of the manuscript are very similar to the previous manuscript, with some differences by distinguishing three latitudinal bands"*. Our conclusions for the near global mean are indeed similar to the previous manuscript, which is why we give more attention to the high latitudes that were not studied in the previous manuscript. We point out that DNA active irradiance is expected to change differently at high latitudes than at near-global scale after around 2050. It will continue to decline at high latitudes mainly due to ozone recovery (cloud cover changes

are not significant), while it is expected to increase on a near-global scale affected by the reduction of cloud cover from climate change. This opposite behaviour is not a finding of the previous study, and we believe it is worth sharing with the scientific community. Of course, it is an outcome that emerges from the simulations of a single climate-chemistry model, and as such, it may well turn out to be true or false. Verification of the results from other model simulations would be useful. It is important to note that our free running simulations were designed according to the definitions for the reference and sensitivity simulations provided by the IGAC and SPARC communities to address emerging science questions, improve process understanding and support upcoming ozone and climate assessments (Eyring et al., 2013).

Reference

Eyring, V., Lamarque, J.-F., Hess, P., Arfeuille, F., Bowman, K., Chipperfield, M., Duncan, B., Fiore, A., Gettelman, A., Giorgetta, M., Granier, C., Hegglin, M., Kinnison, D., Kunze, M., Langematz, U., Luo, B., Martin, R., Matthes, K., Newman, P., Peter, T., Robock, A., Ryerson, A., Saiz-Lopez, A., Salawitch, R., Schultz, M., Shepherd, T., Shindell, D., Stähelin, J., Tegtmeier, S., Thomason, L., Tilmes, S., Vernier, J.-P., Waugh, D., and Young, P.: Overview of IGAC/SPARC Chemistry-Climate Model Initiative (CCMI) Community Simulations in Support of Upcoming Ozone and Climate Assessments, SPARC Newsletter, 40, 48-46, 2013.

*Since the model used in the analysis is a global model, the restriction to 13 specific sites must be for good reason. The only reason I can see is that this allows comparison between the ground based UV stations with the model at these locations, in order to validate the model. I am not convinced by this argument for the following reasons:*

*• The model has a resolution of 3°x3°, which is a huge area for which the point measurement has to be representative for. I doubt that this is the case for several stations where the surrounding area is inhomogeneous, such as mountain tops (Mauna Loa, Sonnblick, Zugspitze), or in valleys (Aosta), or by being in a town with very heterogeneous surroundings (sea, mountains, tropospheric ozone and aerosols such as Athens).*

*• The correlation for DNA weighted UV irradiance is actually not very good, and the figures in the supplementary material show quite different behaviour. A general comment is that the correlation is not the only measure for the agreement between two datasets, but also the slope between two datasets (from a scatter plot), are significantly different from one, showing that the model results disagree quite significantly from the*

*measurements, see Table S1 with a summary of the statistics. For me these comparisons do not support any validation of the model.*

**Answer:**

Yes, the reason for the restriction to specific sites was because it allowed comparison between the ground-based UV stations and the model at these locations. We did not have any better way to test the model simulations with real measurements. We agree with the reviewer's comment that these comparisons do not support any validation of the model, which is why we do not use the word "validation" in the text but the word "evaluation", which is more appropriate.

More specifically, the resolution of the model of 3° x 3° is indeed a large area for which the point measurement has to be representative for. We thank the reviewer for bringing this up, and we have made a note on these comments in the revised Section 3.1, as follows:

"We note here that the model has a resolution of about 3° x 3°, which is a large area for which the point measurement has to be representative for. As such, for stations where the surrounding area is inhomogeneous, such as mountain tops (Mauna Loa, Sonnblick, Zugspitze), or in valleys (Aosta), or by being in a town with very heterogeneous surroundings (sea, mountains, tropospheric ozone and aerosols such as Athens), the model simulations of DNA active irradiance are not expected to be fully representative of the specific UV sites. Thus, the correlation between modelled and measured DNA weighted UV irradiance is not very good at some stations, as shown in the figures provided in the supplementary material. For the same reason, the slope between two datasets can deviate significantly from unity (see Supplement Table S1). Therefore, the comparisons at the individual stations provide a qualitative evaluation of the model's variability, but cannot be considered as a strict validation of the model."

*• The comparisons were performed for past to present data, using a model with prescribed dynamics. However the results of the manuscript are obtained using a free-running model, which as the authors write themselves, has serious shortcomings. The Appendix A discusses this fact, which is appreciated.*

**Answer:**

Indeed, the results of the manuscript were obtained using a free-running model, which has shortcomings during the period of the observed ozone depletion as discussed in

Appendix A. Nevertheless, after the 1980's the model seems to reproduce quite well the observed ozone variability.

*• ozone and Cloud cover are obtained from satellite measurements, which also give a global product.*

**Answer:**

We now analyse the global products of ozone and Cloud cover from satellite measurements and compare them with the respective model simulations. We have added a new figure in the Supplement, as Figure S1, showing the comparisons for ozone and Cloud cover using the zonally averaged data. The results of the statistical comparisons are in line with those from the station averaged data.

[Figure]

(a)                                          (b)

**Figure S1. (a)** Comparison of zonally averaged ozone column from model simulations and satellite measurements for the northern high latitudes (55º–75º N) (upper panel), the near-global mean (50º N–50º S) (middle panel) and the southern high latitudes (55º–75º S) (lower panel). **(b)** Same as **(a)** but for zonally averaged cloud cover. The y-axes show monthly de-seasonalized anomalies (in %) relative to the long-term monthly mean (2000–2018). Shown are monthly anomalies from March to September for the northern high latitudes, and from September to March for the southern high latitudes. For 50º N–50º S, we present all months.

*• The authors themselves mention that some stations might not be very representative, being close to the shore (Barrow and Palmer, line 493).*

**Answer:**

We have added one more station in the northern high latitudes, namely Sodankylä in Finland. The UV irradiance data for Sodankylä were provided by Dr. Kaisa Lakkala and cover the period 1990-2021. The analysis of the model data for Sodankylä was performed by Dr. Kostas Douvis from the Academy of Athens. These two scientists have been added in the list of co-authors. The figures and tables referring to the average of northern high latitude stations have been updated.

*Therefore the benefit of restricting the analysis of the model results to only 13 point locations does not compensate for the results obtained if the model results were analysed as a whole, for example in latitudinal bands, or by selecting specific regions where future changes are expected to be very different (Europe versus Asia, Sahara, …).*

**Answer:**

We now complement the analysis presented in Section 3.2 with zonally averaged data, in order not to restrict the analysis of the model results to only 13 locations according to the reviewer's comment, but to analyse model results for example in latitudinal bands. We have added three new figures in the Supplement that show the changes from the free running simulations for the near global mean, the northern and southern high latitudes based on latitudinal averages. The three new figures are presented below. The results from the analysis of averaging the model data in latitudinal bands are in the same direction with that from the station averages. The results are discussed in Section 3.2. The new figures based on zonally averaged data for the near global mean, the northern and southern high latitudes are the Supplement Figures S2, S3 and S4, respectively.

[Figure]

**Figure S2.** Changes in zonal mean total ozone, zonal mean DNA active irradiance and zonal mean cloud cover for the near global mean (50º N – 50º S), based on simulations with increasing and fixed GHGs mixing ratios. **(a)** REF is the simulation with increasing GHGs according to RCP-6.0. **(b)** FIX is the simulation with fixed GHGs emissions at 1960 levels. **(c)** Difference between the two model simulations, indicating the impact of increasing GHGs. The y-axes in (a) and (b) show yearly averaged data (in %) calculated from de-seasonalized monthly data. The monthly data were de-seasonalized relative to the long-term monthly mean (1990–2019) and were expressed in %. For 50º N–50º S we used all months to calculate the annual average.

[Figure]

**Figure S3.** Same as Figure S2 but for northern high latitudes (55º – 75º N). The y-axes in (a) and (b) show yearly averaged data (in %) calculated from de-seasonalized monthly data. For the northern high latitudes, the annual average refers to the average of monthly anomalies from March to September.

[Figure]

**Figure S4.** Same as Figure S2 but for southern high latitudes (55° – 75° S). The y-axes in (a) and (b) show yearly averaged data (in %) calculated from de-seasonalized monthly data. For the southern high latitudes, the annual average refers to the average of monthly anomalies from September to March.

*Some Specific comments*

*– The simulations of the future climate did not take into account possible solar variabilities (grand minimum, as discussed in Anet, J., Rozanov, S. Muthers, et al., Impact of a potential 21st century "grand solar minimum" on surface temperatures and stratospheric ozone, Geophys. Res. Lett., 40, 4420–4425, doi:10.1002/grl.50806, 2013, and Arsenovic, P., Rozanov, J. Anet, et al., Implications of potential future grand solar minimum for ozone layer and climate, Atmos. Chem. Phys., 18, 3469–3483, doi: 10.5194/acp-18-3469-2018, 2018.*

**Answer:**

The simulations of the future climate were setup to account for the solar variability according to Jöckel et al. (2016), as follows:

"The future solar forcing, used for the projections, has been prepared according to the solar forcing used for CMIP5 simulation of HadGEM2-ES, where the SSTs and SICs are taken from Jones et al. (2011; see also Sect. 3.3 of Jöckel et al., 2016). It consists of repetitions of an idealized solar cycle connected to the observed time series in July 2008. This has been applied consistently for all projections with prescribed SSTs (RC2-base) and the same holds also for the SC-simulations. Here, we deviate from the CCMI recommendations consisting of a sequence of the last four solar cycles (20–23) (see Sect. 3.4 of Jöckel et al., 2016)."

We now explain this in the revised Section 2.3.

New reference added in the References:

Jones, C. D., Hughes, J. K., Bellouin, N., Hardiman, S. C., Jones, G. S., Knight, J., Liddicoat, S., O'Connor, F. M., Andres, R. J., Bell, C., Boo, K.-O., Bozzo, A., Butchart, N., Cadule, P., Corbin, K. D., Doutriaux-Boucher, M., Friedlingstein, P., Gornall, J., Gray, L., Halloran, P. R., Hurtt, G., Ingram, W. J., Lamarque, J.-F., Law, R. M., Meinshausen, M., Osprey, S., Palin, E. J., Parsons Chini, L., Raddatz, T., Sanderson, M. G., Sellar, A. A., Schurer, A., Valdes, P., Wood, N., Woodward, S., Yoshioka, M., and Zerroukat, M.: The HadGEM2-ES implementation of CMIP5 centennial simulations, Geosci. Model Dev., 4, 543–570, https://doi.org/10.5194/gmd-4-543-2011, 2011.

*– Future changes in aerosol loading are expected to be significant in some areas of the globe, having a strong impact on the UV radiation reaching the surface.*

**Answer:**

We agree that future changes in aerosol loading will have a strong impact on the UV radiation reaching the surface. In all simulations analyzed here, we used prescribed aerosol distributions. Since the aerosol distributions have been prescribed, there is no aerosol output for these simulations that we could use to examine the impact of aerosols on the UV radiation.

In the revised Section 2.3, we clarify that:

"In all simulations analyzed here, we used prescribed aerosol distributions. The prescribed aerosol effects are separated into the aerosol surface area, representing chemical effects via heterogeneous chemistry, and the radiative properties influencing the radiation budget (Sect. 3.7 of Jöckel et al., 2016). Due to a glitch, the stratospheric volcanic aerosol was not considered correctly in the free running simulations (Sect. 3.12.1 of Jöckel et al., 2016). Therefore, the dynamical effects of large volcanic eruptions (e.g. Mt. Pinatubo 1991; El Chichón 1982) are essentially not represented in the simulations, except for the contribution to the tropospheric temperature signal induced by the prescribed SSTs. For the specified dynamics simulations, however, this has been corrected. Since the aerosol distributions have been prescribed, there is no aerosol output for these simulations that we could look at. As such, we cannot investigate the impact of future changes in aerosol loading on the UV radiation reaching the surface."

*– The significance of the results are mainly described by correlation coefficient and p values. The supporting figures, for example Figure 1, however show that the variability between the model and ground based stations is very large. Even though scatter plots are also not the method of choice, they would give a better indication how two datasets would scatter, and the slope and associated fitting uncertainties would give an indication on how well the two datasets agree. I would have preferred the authors to have used other metrics as well, such as uncertainties (at the 95% confidence level) derived from the statistical models.*

**Answer:**

The reviewer wanted to see scatter plots and describe the significance of the results with more statistical parameters. We have added the scatter plots, as suggested by the

reviewer, in a new figure. The new figure is Figure 2. The regression lines have been added. The related statistics (slope, error of slope and root mean square error) are now discussed in the text. Section 3.1 has been revised.

[Figure]

**Figure 2.** Scatter plots of DNA active irradiance from simulated and ground-based data shown in Figure 1 for **(a)** 3 UV stations in the northern high latitudes (>55° N), **(b)** 13 UV stations from 50° N to 50° S and **(c)** 4 UV stations in the southern high latitudes (>55° S).

*− The statistical approach of using a MLR technique is interesting, but why did not the authors include in equation 7 also the surface albedo, instead of treating it separately in the following section?*

**Answer:**

The reason for treating it separately in the following section is because we did not analyse surface albedo for all stations but only for Barrow and Palmer.

In the revised manuscript, we have applied equation 7 to the zonal means of 55° – 75° N and 55° – 75° S, also including the surface albedo parameter. The results are included in Section 3.4, as follows:

"In order to better represent the northern and southern high latitudes, we also applied the MLR model to the large-scale zonal means of 55° – 75° N and 55° – 75° S. For the northern high latitude zone, the findings are in the same direction as those found for Barrow. We estimate that ~31% of the DNA active irradiance trend is determined by the trend in ozone, and that ~14% and ~32% of the DNA active radiation trend are explained by trends in clouds and surface albedo, respectively. For the southern high latitude zone, we estimate that the largest part of the DNA active irradiance trend is determined by the trend in ozone, and that the contributions of cloud and albedo trends are small."

*− The datasets would have been ideally suited to be analysed using the very powerful Dynamical Linear Modelling (DLR), for example, Alsing, (2019) dlmmc: Dynamical linear model regression for atmospheric time-series analysis. Journal of Open Source Software, 4(37), 1157, https://doi.org/10.21105/joss.01157, and Laine, M., Latva-Pukkila, N., and Kyrölä, E.: Analysing time-varying trends in stratospheric ozone time series using the state space approach, Atmos. Chem. Phys., 14, 9707–9725, https://doi.org/10.5194/acp-14-9707-2014, 2014.*

**Answer:**

Yes, they would, but we do not have experience with the Dynamical Linear Modelling. We leave this kind of analysis for a future study.

*− Figure 1: 50N-50S, there is a striking difference between model and measurements around the year 2012, of about 15%, which seem not be seen in either ozone or cloud cover. Did the authors investigate this feature?*

**Answer:**

We thank the reviewer for noticing this. By mistake we had plotted the wrong line from the model. We now plot the new correct line, and the striking difference is not there anymore. Figure 1 and related statistics in Table 2a have been corrected. The corrected Figure 1 is as follows:

[Figure]

**Figure 1.** Comparison of model simulations of DNA active irradiance with averages of ground-based measurements at 3 UV stations in the northern high latitudes (>55º N) (upper panel), 13 UV stations from 50º N to 50º S (middle panel) and 4 UV stations in the southern high latitudes (>55º S) (lower panel). The y-axis shows monthly de-seasonalized DNA active irradiance data (in %). The monthly data at each station were de-seasonalized by subtracting the long-term monthly mean (2000–2018) pertaining to the same calendar month and were expressed in %. Then, the average over each geographical zone was estimated by averaging the de-seasonalized data of the stations belonging to each geographical zone. Shown are data from March to September for the northern high latitudes and from September to March for the southern high latitudes.

*− In the figures from the supplement, the model variabilities of the DNA weighted irradiance are much larger than the corresponding measurements for : Barrow, Villeneuve d'Ascq, Aosta, Lauder, Ushuaia, while they are in better agreement for Summit, Thessaloniki, Boulder, Mauna Loa, or Alice Springs. For Athens, between approx. 2012 and 2015 the measurements are significantly higher than the model results, why is that so?*

**Answer:**

As it is now clarified in the revised Section 3.1, the main reason for these differences in the variability of the DNA weighted irradiances is that we compare averages for 3° x 3° pixels (model simulations) with measurements performed at specific sites (representing narrow areas in the corresponding pixels). Thus, environmental features within the model pixel – that do not affect the station where measurements are performed – may lead to increased variability for the model with respect to the station. For example, the sites of Aosta, Lauder and Ushuaia are surrounded by very high mountains where surface albedo varies significantly in the year and affects strongly the levels of UV irradiance. The measurement sites however are at lower altitudes and changes in surface albedo do not affect strongly the levels of surface UV irradiance. In other sites (e.g., Villeneuve d'Ascq) environmental conditions are possibly less variable with respect to the average conditions in the pixel wherein they belong. Other sites (e.g., Summit, Thessaloniki, Boulder, Mauna Loa, or Alice Springs) are possibly more representative for the average conditions in the pixels wherein they belong. For Athens, the most possible explanation for the much higher measured UV relative to the UV simulated by the model in 2011 – 2014 is again that the model cannot accurately capture changes in air quality (e.g., aerosols, tropospheric ozone etc.) at the city since it represents a much wider area. We were not able to find aerosol optical properties in the UV for the same period for the site in order to verify our assumption but we intend to further investigate these differences in the future.

We make a note on these comments in the updated Supplement of this study.

*– The cloud cover from Modis/Terra and the model show no correlation for most stations, apart for example for Aosta, which is slightly better. Can the authors provide some comments why some stations show better agreement than others?*

**Answer:**

It would be nice if we obtained good model – satellite correlations from all datasets. We would have perfect model simulations and perfect satellite measurements at all locations. Frankly speaking, we cannot say which of the two datasets is responsible for the smaller agreement at some stations or if both are. But statistically speaking, we find that the majority of the stations (13 of 20 stations) show medium to good correlations (between 0.5 and 0.7), 5 stations show small to medium correlations (between 0.3 and 0.5), and only 2 stations show no correlation. The stations that show no correlation are Summit and South Pole. Both stations are high-altitude sites located at high latitudes with year-round snow cover and albedos of larger than 0.95. Multiple scattering

between the surface and clouds greatly reduces cloud effects (Nichol et al., 2003). Mauna Loa (MLO) is also a high-altitude site. MLO is interesting, not only because it is also at high altitude, but because there are often clouds below the station, which enhance downwelling radiation similar to the effect of high albedo. Both in Antarctica and MLO, UV radiation is scattered up either by snow or a cloud layer, and then Rayleigh-scattered down to increase downwelling irradiance. However, we find that the correlation is 0.592 for the cloud case in MLO, suggesting that other contributing processes might account for somewhat successful cloud simulations in the EMAC model which would require further investigation. The stations with small to medium correlations (between 0.3 and 0.5) are Haute Provence, Athens, Lauder, Ushuaia and Palmer. We remind that all correlations were derived from de-seasonalized data, i.e. data after removing the mean seasonal cycle of the period 2000-2018. This was performed because we wanted to evaluate the long-term variability of cloud cover and not its seasonal cycle.

We make a note on these comments in the updated Supplement of this study.

Reference

Nichol, S. E., Pfister, G., Bodeker, G. E., McKenzie, R. L., Wood, S. W., and Bernhard, G.: Moderation of cloud reduction of UV in the Antarctic due to high surface albedo, J. Applied Meteorology, 42, 1174–1183, DOI: https://doi.org/10.1175/1520-0450(2003)042<1174:MOCROU>2.0.CO;2, 2003.

**Reply to Reviewer #2**

*The manuscript presents valid and useful analysis with adequate source data, solid statistical analysis and interpretation and reasonable (but not spectacular or unexpected) conclusions. I am favourable on the scientific merits of this work despite the fact that a very similar analysis (now enriched with more data here) has already been published before by the same first author (I do not see this as an obstacle for publication). I only have the following comments which I would like to see addressed (numbers indicate the respective manuscript lines):*

**Answer:**

We would like to thank the reviewer for the useful comments and for finding our work worthy of publication despite of our previously published paper. All comments have been answered and the text has been revised accordingly.

*GENERAL*

*1-3: The title oversells the role of GHGs. The analysis of UV changes is done for simulations with and without time-varying GHGs and without doubt, modelled changes (=increases) of GHGs are driving the UV changes. But the actual UV change is mainly brought about by cloud changes (driven by changes in GHGs) that correctly the manuscript places in primary focus.*

**Answer:**

The reviewer correctly writes that the actual UV change is mainly brought about by cloud changes (driven by changes in GHGs). We have revised the title of the paper to also mention the cloud changes, as follows:

"Ozone, DNA active UV radiation and cloud changes for the near global mean and at high latitudes due to enhanced greenhouse gas concentrations".

*100-109: In direct relation to my comment for lines 1-3 and perhaps to justify the prominent insertion of the GHGs in the title, 1-2 additional lines should elaborate on why/how the GHG changes drive the cloud changes (that actually effect the UV changes).*

**Answer:**

We now discuss why/how the GHG changes can drive the cloud changes, as follows:

"Norris et al. (2016) provided evidence for climate change in the satellite cloud record. They estimated fewer clouds over the mid-latitudes from 1983 to 2009 and concluded that the observed and simulated cloud change patterns are consistent with poleward retreat of mid-latitude storm tracks, expansion of subtropical dry zones, and increasing height of the highest cloud tops at all latitudes. The primary drivers for these changes were found to be the increasing GHG concentrations and a recovery from volcanic radiative cooling (Norris et al., 2016). In the same direction, Scheider et al. (2019) showed that stratocumulus clouds, some of the planet's most effective cooling systems, become unstable and break up into scattered clouds under increasing GHG concentrations. Their results also showed that less clouds will trigger additional surface warming in addition to that from the rising $CO_2$ levels (Scheider et al., 2019). Both studies provided indications that increasing GHGs can affect clouds, which in turn can affect the UV radiation changes."

Two references have been added:

Norris, J. R., Allen, R. J., Evan, A. T., Zelinka, M. D., O'Dell, C. W., and Klein, S. A.: Evidence for climate change in the satellite cloud record, Nature, 536, 72–75, doi: 10.1038/nature18273, 2016.

Schneider, T., Kaul, C. M., and Pressel, K. G.: Possible climate transitions from breakup of stratocumulus decks under greenhouse warming, Nature Geoscience, 12, 163–167, https://doi.org/10.1038/s41561-019-0310-1, 2019.

*177-178: From the way it is written, I deduce that the reference simulation includes additional 10 years of run for spin-up, while the sensitivity one, no. Is this correct? If yes, does this affect the ozone simulation? Ideally, shouldn't the runs be identical (with only difference the time-varying GHGs)?*

**Answer:**

Indeed, the reference simulation (RC2-base-04) includes 10 year spin-up (1950-1960) to overcome the original initial conditions chosen for 1950, in particular the initialized distributions of the chemical compounds, in order to get the model on the RCP-6.0 track. From 1960 onwards, both simulations (the reference and the sensitivity SC2-

fGHG-01) both follow the RCP-6.0 scenario with the GHG kept at 1960 levels for the sensitivity simulation.

In other words, SC2-fGHG-01 was "branched off" from the reference and both share the same spin-up period.

As such, the only difference from 1960 to 2100 is indeed the time-varying vs. constant GHG.

*267-268: The importance of this comment goes beyond technicalities so I insert it here. Currently the manuscript, throughout the figures, labels the three model runs according to their original names given by the modellers for specific reasons but are not necessary for the journal paper reader (in contrast they make harder following the figure content). The model run labels in the figure must be short and intuitive, for example:*
*SC1SD-Base02 -> HIS (for historical/hindcast)*
*RC2-Base04 -> SCE (for time-varying GHGs)*
*SC2-fGHG-01 -> FIX (for fixed GHGs)*

**Answer:**

The notation of our simulations has been used in previous publications as well, and it largely follows the CCMI notation: R stands for Reference, C for CCMI phase 1, SD for specified dynamics (i.e. nudged), and base for "basic setup" vs. anything else, for instance fGHG for fixed Greenhouse Gases. The number determines either the correct realisation after n-1 attempts or the ensemble member.

In particular "SD" is an important indicator for the setup, because it distinguishes specified dynamics from free running simulations.

Since this notation has been used in other publications as well, we are hesitating to change them; however, since the reviewer thinks it is hard for the reader to follow the figure content, we will change the notation as follows:

In the description we once introduce "our" names and later use the suggested abbreviations, for instance:

- SC1SD-base-02 --> further abbreviated as HIS-SD for historical specified dynamics
- RC2-base-04 --> further denoted as REF for reference, and
- SC2-fGHG-01 --> further denoted as FIX for fixed GHGs.

*590-592: how many Sigmas is this uncertainty range defined for? please clarify.*

**Answer:**

We have revised the lines which now mention the Sigmas as follows:

"We conducted a separate analysis on total cloud cover variability and trends through the 21$^{st}$ century, using the available simulations from the CCMI-1 REF-C2 set, which showed that the EMAC CCM results fall well within the range of uncertainty (i.e., ± 2σ), and is close to the ensemble average (± 1σ)."

*TECHNICAL*

*176: Move the description of the runs in a different paragraph for easier reading.*

**Answer:** Done

*179-185: " Furthermore, we have analyzed the ...". The description of the EMAC RC1SD-base-10 can preceed the description of the scenario runs.*

**Answer:** Done

*186-192: Further description of the scenario runs must be merged with the previous one in lines 176-179.*

**Answer:** Done

*211-225: The deseasonalisation definitions and the t-test formula for the correlation coefficient formula may be introduced as a "statistical methods (or formulas)" sub-section in a Data and Methods Section.*

**Answer:** Done. We have moved the lines to a new Section 2.4 "Statistical methods".

*256: replace "and of the parameters" with "and the parameters"*

**Answer:** Done

*408-423: the mathematics (equations etc) used for the statistical tests for difference may be introduced as a "statistical methods (or formulas)" sub-section (same as in comment for lines 211-225) in a Data and Methods Section.*

**Answer**: Done. We have moved the lines to a new Section 2.4 "Statistical methods".

---

## Author Response (AR2)

Reply to the Editor

We would like to thank the editor for his positive decision. We have read the document once more and corrected some of the sentences on the new/added text, as recommended.

The corrections on the new/added text are attached below with blue colour.

With kind regards,

Kostas Eleftheratos

[revised manuscript text omitted]
 | −0.21 ± 0.30 | −0.10 ± 0.23 | 0.32 ± 0.20 | −0.20 ± 0.43 | 0.02 ± 0.25 | 0.19 ± 0.27 | −0.29 ± 0.45 | −0.39 ± 0.25 | 0.29 ± 0.27 |

[revised manuscript text omitted]